# hu.MAP 2.0: integration of over 15,000 proteomic experiments builds a global compendium of human multiprotein assemblies

Kevin Drew[*,†] (iD), John B Wallingford & Edward M Marcotte[**] (iD)

## Abstract

**A general principle of biology is the self-assembly of proteins into functional complexes. Characterizing their composition is, therefore, required for our understanding of cellular functions. Unfortunately, we lack knowledge of the comprehensive set of identities of protein complexes in human cells. To address this gap, we developed a machine learning framework to identify protein complexes in over 15,000 mass spectrometry experiments which resulted in the identification of nearly 7,000 physical assemblies. We show our resource, hu.MAP 2.0, is more accurate and comprehensive than previous state of the art high-throughput protein complex resources and gives rise to many new hypotheses, including for 274 completely uncharacterized proteins. Further, we identify 253 promiscuous proteins that participate in multiple complexes pointing to possible moonlighting roles. We have made hu.MAP 2.0 easily searchable in a web interface (http://humap2.proteincomplexes.org/), which will be a valuable resource for researchers across a broad range of interests including systems biology, structural biology, and molecular explanations of disease.**

**Keywords** data integration; human protein complexes; mass spectrometry; moonlighting proteins

**Subject Categories** Methods & Resources; Post-translational Modifications & Proteolysis

**Mol Syst Biol. (2021) 17: e10016**

## Introduction

Macromolecular protein complexes carry out a wide variety of functions in the cell including essential functions such as replication, transcription, translation, and protein degradation (e.g., MCM-ORC, RNA polymerase, ribosome, proteasome) (Alberts, 1998; Hartwell *et al*, 1999). The disruption of protein complexes is implicated in many human diseases (Goh *et al*, 2007; Bruel *et al*, 2017) and many

therapeutics target protein complexes (Arkin *et al*, 2014). The formation of protein complexes to carry out biological function is a general principle of biology and characterizing their composition is therefore a basic requirement to our full understanding of cellular functions. Unfortunately, we still lack knowledge of a comprehensive set of protein complexes for the human cell.

To address this gap in knowledge, there is an ongoing worldwide effort to identify all protein interactions in human cells. High-throughput protein interaction screens using affinity purification (AP-MS) (Malovannaya *et al*, 2011; Hein *et al*, 2015; Huttlin *et al*, 2015, 2017) have greatly increased coverage of protein interactions across the proteome. Likewise, biochemical separation strategies, such as co-fractionation mass spectrometry (CF-MS), have provided orthogonal approaches to identifying protein complexes (Havugimana *et al*, 2012; Kristensen *et al*, 2012; Kirkwood *et al*, 2013; Wan *et al*, 2015). Although these methods have identified tens of thousands of protein interactions, they still have limited coverage of the entire human interactome.

Fortunately, these high-throughput methods are orthogonal, each sampling different parts of the human proteome and identifying sets of interactions which do not completely overlap. We previously re-analyzed and integrated three of the largest datasets available at the time (Hein *et al*, 2015; Huttlin *et al*, 2015; Wan *et al*, 2015), over 9,000 mass spectrometry experiments, to build a more complete and accurate set of protein complexes (Drew *et al*, 2017). Our resource, hu.MAP, identified interactions for over a third of all human proteins.

We envision interactomes as evolving entities, growing, and improving as new technologies and datasets emerge. Here, we introduce hu.MAP 2.0, which we find to be the most accurate and comprehensive human protein complex map available to date. hu.MAP 2.0 is an integration of over 15,000 mass spectrometry experiments and identifies 6,965 complexes consisting of 57,148 unique interactions among 9,963 human proteins. Multiple performance metrics demonstrate that hu.MAP 2.0 outperforms our previous map and other complex maps available in the literature. We further show that complexes in hu.MAP 2.0 are not only highly enriched for specific literature-curated annotations but also have

Department of Molecular Biosciences, Center for Systems and Synthetic Biology, University of Texas, Austin, TX, USA
*Corresponding author. Tel: +1 312 996 9341; E-mail: ksdrew@uic.edu
**Corresponding author. Tel: +1 512 471 5435; E-mail: marcotte@utexas.edu
†Present address: Department of Biological Sciences, University of Illinois at Chicago, Chicago, IL, USA

greater coverage of completely uncharacterized genes. Finally, we highlight several new biological findings that illustrate the utility of hu.MAP 2.0 as a resource for biological discovery.

# Results

Our strategy in building a comprehensive map of protein complexes involves the integration of the many orthogonal experimental protein interaction datasets available using a custom machine learning pipeline as shown in Fig 1A. Each individual experimental dataset identifies different sets of protein interactions and therefore combining them results in a more accurate and comprehensive set of interactions. Our pipeline combines quantified features from these datasets using a support vector machine (SVM) classifier which calculates a confidence score of two proteins interacting. This results in a large protein interaction network. The network is subsequently searched for dense regions of highly connected proteins which represent individual complexes. The identified complexes are ranked by a clustering confidence value. We call the resulting set of complexes, hu.MAP 2.0. The map of complexes contains many known complexes such as the EIF2B complex, Spliceosome, RNA Pol III, and IFT-A complex (Fig 1A Positive Control Examples) as well as many novel insights into the physical biology of the cell.

**Integration of over 15,000 mass spectrometry experiments**

To construct hu.MAP 2.0, we integrated over 15,000 previously published mass spectrometry experiments using our custom machine learning framework. We built upon the 9,000 mass spectrometry experiments used for hu.MAP 1.0 (Hein *et al*, 2015; Huttlin *et al*, 2015; Wan *et al*, 2015; Drew *et al*, 2017) by incorporating additional affinity purification data from Bioplex 2 (Huttlin *et al*, 2017) and (Boldt *et al*, 2016) as well as proximity labeling data from (Gupta *et al*, 2015) and (Youn *et al*, 2018) (Fig 1A). Our rationale for including these datasets was twofold. First, each dataset samples a different set of bait proteins which provides increased coverage of the interactome. Second, the methods are orthogonal and complementary, where affinity purification targets stable interactions, and the proximity labeling datasets potentially also capture transient *in vivo* interactions. The "Upset" plot (Lex *et al*, 2014) in Fig 1B shows that tens of thousands of protein pairs are represented by 2 or more datasets providing orthogonal evidence for those interactions. This greatly enhances our framework's ability to identify true interactions from false ones.

Additionally, we applied our Weighted Matrix Model (WMM) technique which we previously demonstrated identifies many new high confidence interactions from affinity purification data (Hart *et al*, 2007; Drew *et al*, 2017). Contrary to the traditional spoke and matrix models used to interpret AP-MS data that only consider one pull-down experiment at a time, the WMM technique takes all pull-down experiments from a given dataset into account and determines protein pairs that are in the same pull-down experiments more often than random chance. The WMM balances both the false-negative and false-positive issues that face both the spoke and matrix models and therefore is capable of identifying novel interactions. More specifically, since a spoke model only considers interactions between a bait protein and a prey protein, all true interactions between prey proteins are missed leading to high false-negative rates for the spoke model. Alternatively, a naive matrix model does consider interactions between prey proteins limiting false negatives but does so by treating all prey pairs equally. Some of these prey pairs will participate in the same complex but since proteins participate in multiple complexes, two prey proteins pulled down by the same bait are not guaranteed to interact. This leads to a high degree of false positives for the naive matrix model. The WMM considers all prey pairs as interactors but weights them according to the frequency they occur together while controlling for "frequent flyer" or "sticky" proteins. Therefore, by considering all prey pairs the WMM has better false-negative rates than the spoke model, and by accurately measuring the specificity of the prey pairs, the WMM has better false-positive rates than the naive matrix model. The WMM can be applied to proximity labeling experiments and also, in contrast to the traditional models, the WMM can be applied to datasets that were not exclusively collected for the purpose of identifying protein interactions. Specifically, we applied our WMM to > 3,000 RNA hairpin pull-down experiments (Treiber *et al*, 2017) and incorporated the results into our framework. Figure 1B shows WMM overlaps substantially with other methods but also provides additional information for many pairs of proteins not covered in the other datasets (> $2.2 \times 10E6$ protein pairs attributed to WMM alone).

**hu.MAP 2.0 network is highly accurate**

We next trained an SVM classifier to determine whether two proteins interact in a macromolecular complex. The classifier uses 292 features computed from the mass spectrometry experiments and is trained on examples consisting of co-complex protein interactions from a set of > 1,100 literature-curated CORUM complexes (Giurgiu

---

**Figure 1. Machine learning framework to identify protein complexes.**

A Graphical description of computational pipeline to integrate > 15,000 mass spectrometry experiments. Number of experiments used is listed next to each technique (see also Table 1). A Support Vector Machine (SVM) classifier was trained using numerical measures (i.e., features) on pairs of proteins calculated from original mass spectrometry data and training labels from literature-curated complexes (CORUM). The classifier was then used to construct a protein interaction network by calculating a confidence score for all pairs of proteins for their propensity to interact. Clustering parameters were then learned from training complexes, and five final sets of clusters were chosen ranked in order of confidence from "Extremely High" to "Medium". The union of these selected clusterings represents the final set of hu.MAP 2.0 complexes. Networks of previously known protein complexes identified by this pipeline which were not in the training set of complexes are shown as positive control examples.

B "UpSet" plot (Lex *et al*, 2014) displaying the intersections of protein pairs for all integrated datasets. Each set of connected black dots represents the intersection of the respective datasets. Vertical bar plot displays protein pair count of intersection. Light gray dots are datasets not included in the intersection. Single unconnected black dots represent protein pairs that are only present in a single dataset. Horizontal bar plot represents total protein pair count in each dataset. The plot shows the Weighted Matrix Model (single black WMM dot) provides additional information for many pairs of proteins (> $2.2 \times 10E6$) that would be limited otherwise.

   

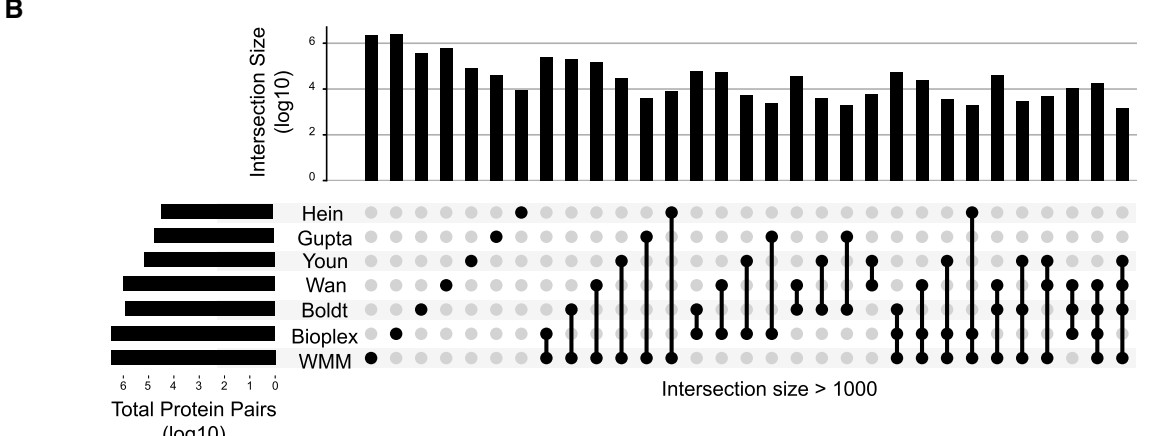

Figure 1.

et al, 2019). All features, with the exception of WMM features, were downloaded from the original publications (see Materials and Methods for a complete list). The set of CORUM complexes were split into two equal proportion non-overlapping test and train subsets. We then derived sets of co-complex interactions from these test and train complex subsets. The classifier was parameterized using 5× cross-validation on the train co-complex interactions and applied to 17.5 million pairs of proteins scoring each one for their ability to interact based on the input data (see Materials and Methods). The full list of scored pairs and the complete feature matrix can be found in the Data availability section.

To evaluate the performance of our SVM classifier, we examined the confidence scores of 8,337 co-complex interactions in our leave-out test set using a Precision-Recall framework (Fig 2A). Precision-Recall frameworks are the preferred method of evaluating imbalanced binary classification problems such as protein interaction identification (Davis & Goadrich, 2006; Saito & Rehmsmeier, 2015). We see that hu.MAP 2.0 outperforms our previous complex map, hu.MAP 1.0, as well as all other datasets including Bioplex 2, Wan et al and Hein et al, demonstrating the power of data integration. For example, a direct comparison of hu.MAP 2.0 to hu.MAP 1.0 at a fixed precision of 0.5 results in an increase of 67% recall relative to hu.MAP 1.0. We also evaluated the increase in total high confidence (precision = 0.9) protein interactions identified by hu.MAP 2.0 and saw an increase of > 1,880 high confidence interactions over hu.MAP 1.0. Additionally, we see a drop in performance when we remove the WMM features from hu.MAP 2.0 further showing how important the WMM features are to performance improvements. Finally, we compare hu.MAP 2.0 to the HuRi dataset which encompasses a 17,500 × 17,500 all-by-all yeast 2-hybrid screen (Luck et al, 2020). Yeast 2-hybrid aims to capture only direct protein-protein interactions and has previously shown good performance on benchmarks of binary interactors (Luck et al, 2020). Here we see HuRi underperforms all other networks when evaluated on co-complex interactions likely due to its inability to identify indirect physical interactions.

Recently, a co-regulation map based on protein expression was shown to capture relationships among proteins that do not necessarily interact or co-localize (Kustatscher et al, 2019). This dataset therefore provides an independent test of the quality of our protein interactions. When we compared the hu.MAP 2.0 interactions to the most co-expressing pairs in Kustatcher et al, we see a highly significant overlap (P-value < 10E−10, hypergeometric test) indicating a high degree of consistency between the orthogonal datasets.

## hu.MAP 2.0 complex map

Once we confirmed our protein interaction network was of high quality, we then clustered this network using a parameterized two-stage clustering algorithm to identify highly connected proteins which represent protein complexes. Briefly, the algorithm first filters the network based on the confidence score from the SVM classifier and then uses the ClusterOne algorithm (Nepusz et al, 2012) to cluster the thresholded network. ClusterOne often produces large clusters containing proteins from multiple complexes. To remedy this, each resulting cluster is further clustered using the MCL algorithm (Enright et al, 2002) (Fig 2B). There are several parameters for each algorithm that require optimization in addition to an SVM

confidence score threshold of the input network. We therefore take an agnostic approach and optimize these parameters by generating clusterings for over 1,700 parameter combinations. Each parameter combination is evaluated using the k-clique precision-recall performance measure (Drew et al, 2017) comparing the resulting clusters from the specific parameter combination with the CORUM-based set of training complexes (Fig 2C). The resulting clusterings vary substantially with regards to their performance but ultimately show a familiar pattern of a trade-off between precision and recall. We therefore selected five clusterings that balance the precision-recall trade-off. For example, clustering 1 (green) is a set of clusters with "extremely high" precision but low recall while clustering 5 (gray) is a set of clusters with "medium" precision yet higher recall (Figs 1A and 2C). This allows us to rank our confidence of a protein complex based on its appearance in one of the five clusterings. We then combined all five selected clusterings into a union set while preserving their precision rank (Dataset EV1).

To evaluate the union of clusterings that represent our final hu.MAP complexes, we again use the k-clique precision-recall performance measure but now calculated on the leave-out set of test complexes. As shown in Fig 2D, our hu.MAP 2.0 complexes balance both precision and recall. Additionally, we see a consistent trend of k-clique precision and recall values for our individual clusterings between both the train (Fig 2C) and test sets (Fig 2D). This suggests the confidence ranking given to each complex is robust. We also observe our final set of complexes outperforms our previous hu.MAP 1.0 complexes and other previous state-of-the-art complex maps. Taken together, this points to hu.MAP 2.0 complexes as being highly accurate and spanning a large portion of all human protein assemblies.

## Identification of multifunctional promiscuous proteins

Now that we have established hu.MAP 2.0 as a highly accurate resource of human protein complexes, we can ask questions about protein assemblies in a systematic way. Specifically, one systematic question we can ask is how prevalent are promiscuous proteins in stable protein assemblies? That is, how often do we see proteins participating in multiple different complexes and presumably performing orthogonal functions (sometimes termed "moonlighting" (Chapple et al, 2015; Jeffery, 2015)). We therefore created a non-redundant set of complexes (see Materials and Methods) and surprisingly identified 253 proteins that participate in multiple complexes (Dataset EV2). This is a 53% increase over the same analysis if done on hu.MAP 1.0.

The majority of promiscuous proteins appear only in two complexes and the largest number of complexes a protein participates in is four, suggesting these are not just "sticky" interactions but rather proteins participating in multiple specific complexes. The 253 proteins constituted nearly 7.5% of proteins in the non-redundant set of complexes. This compares to 106 moonlighting human proteins identified in the MoonProt database (Mani et al, 2015). Additionally, when we compare expression levels from the Human Protein Atlas (Uhlén et al, 2015) across tissues of promiscuous proteins versus non-promiscuous proteins, we see the bulk of the distributions substantially overlaps, suggesting the promiscuous proteins do not appear in a greater number of complexes due to higher expression levels (Fig EV1).

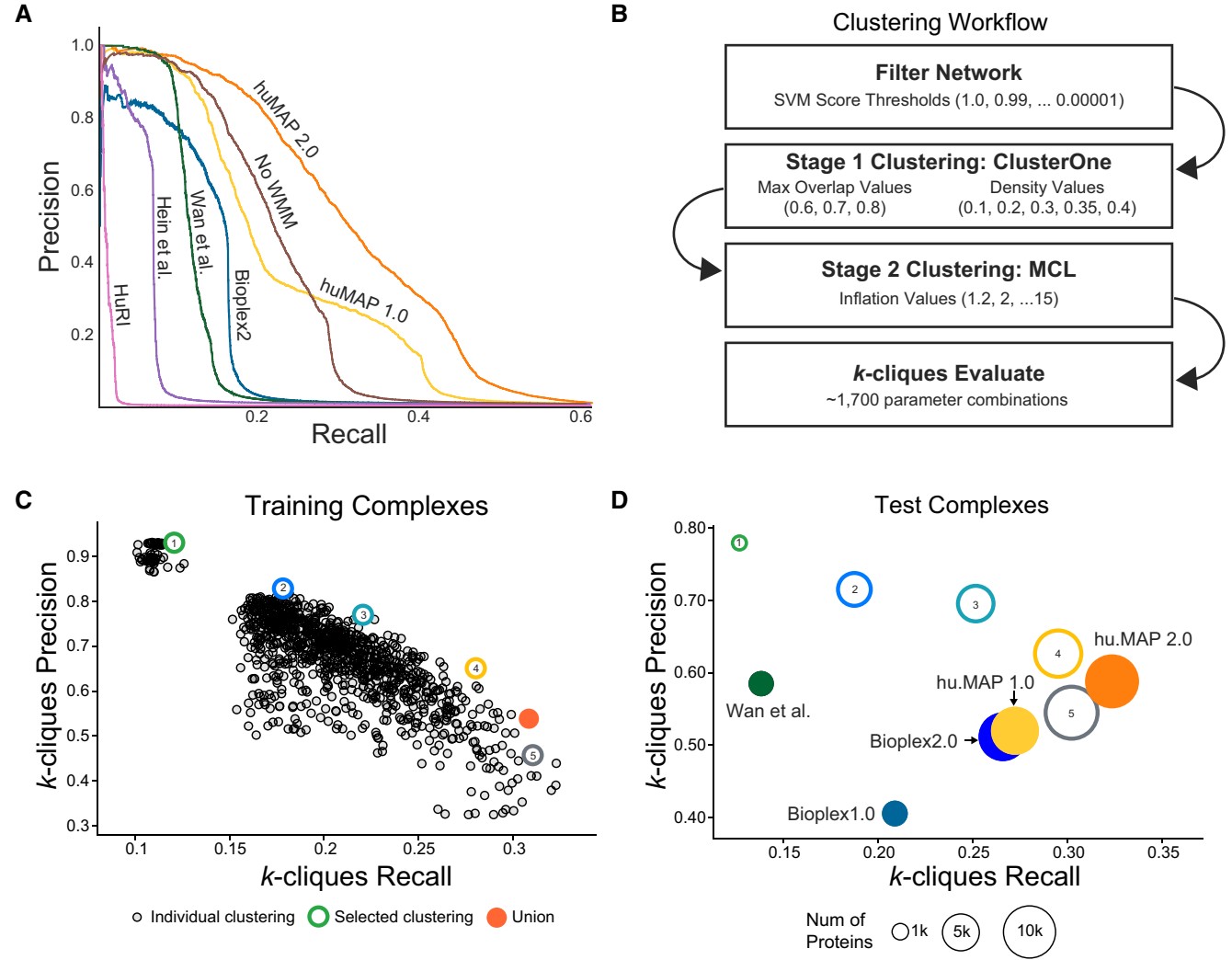

**Figure 2. hu.MAP 2.0 outperforms previous complex maps.**

A Precision-Recall (PR) plot evaluated on a test (leave-out) set of literature-curated co-complex pairwise protein interactions. The plot shows hu.MAP 2.0 is more accurate and comprehensive than previous published datasets. The plot also evaluates the performance of predictions without the Weighted Matrix Model (WMM) and shows the WMM substantially improves performance.

B Clustering workflow used to identify protein complexes in hu.MAP 2.0 protein interaction network. The network is first filtered based on the confidence score produced by the Support Vector Machine (SVM). The filtered network is then clustered using a two-stage approach, clustering first using ClusterOne, and then further clustering with MCL. The resulting clusters are then evaluated using the *k*-clique method (see Materials and Methods) on training complexes. Approximately 1,700 parameter combinations were evaluated, each producing a unique set of clusters, sweeping SVM score filter thresholds, and clustering parameters (i.e., ClusterOne Max Overlap, ClusterOne Density, and MCL Inflation).

C *k*-clique Precision-Recall (*k*PR) scatter plot of 1,700 clustering parameter sets. Five clusterings (colored hollow circles) were selected representing varying degrees of confidence balancing the trade-off between precision and recall. The five selected clusterings were combined as a final set of clusters (orange filled circle).

D *k*PR scatter plot of hu.MAP 2.0 complexes (orange filled circle) and other published complex maps (colored filled circles) evaluated on a test set of literature-curated complexes. hu.MAP 2.0 complexes increase in both precision and recall relative to other maps. Also plotted are the five sets of complexes at different levels of confidence (colored hollow circles) demonstrating consistency between the level of confidence determined from training set (B) and test set.

Figure 3A and B shows an example of a promiscuous protein, HSPA9, participating in two unrelated protein complexes. HSPA9 is a multilocational (mitochondria and nucleus) and multifunctional protein, playing a role in mitochondrial import and stress response (Wadhwa *et al*, 2002). We identify HSPA9 participating in two complexes that reflect its multifunctional role. First, we identify HSPA9 in complex HuMAP2_01130, a heat shock response complex supported by AP-MS experiments (Guruharsha *et al*, 2011; Huttlin *et al*, 2015) and co-fractionation evidence (Wan *et al*, 2015). And second, we identify HSPA9 in complex HuMAP2_00358, a mitochondrial protein import complex supported by AP-MS experiments (Malovannaya *et al*, 2011; Huttlin *et al*, 2015; Boldt *et al*, 2016), co-fractionation evidence (Wan *et al*, 2015), and WMM based on (Treiber *et al*, 2017). To further verify HSPA9's membership in these

two complexes, we inspected sparkline traces of two co-fractionation experiments (Wan *et al*, 2015) (Fig 3C). We identified two separate elution peaks of HSPA9 which correspond to the two complexes. This example shows the ability of our complex map to identify multifunctional promiscuous proteins and place them into their respective non-overlapping functional complexes.

Saeed and Deane (2006) previously showed a strong relationship between protein age and interaction connectivity. We therefore

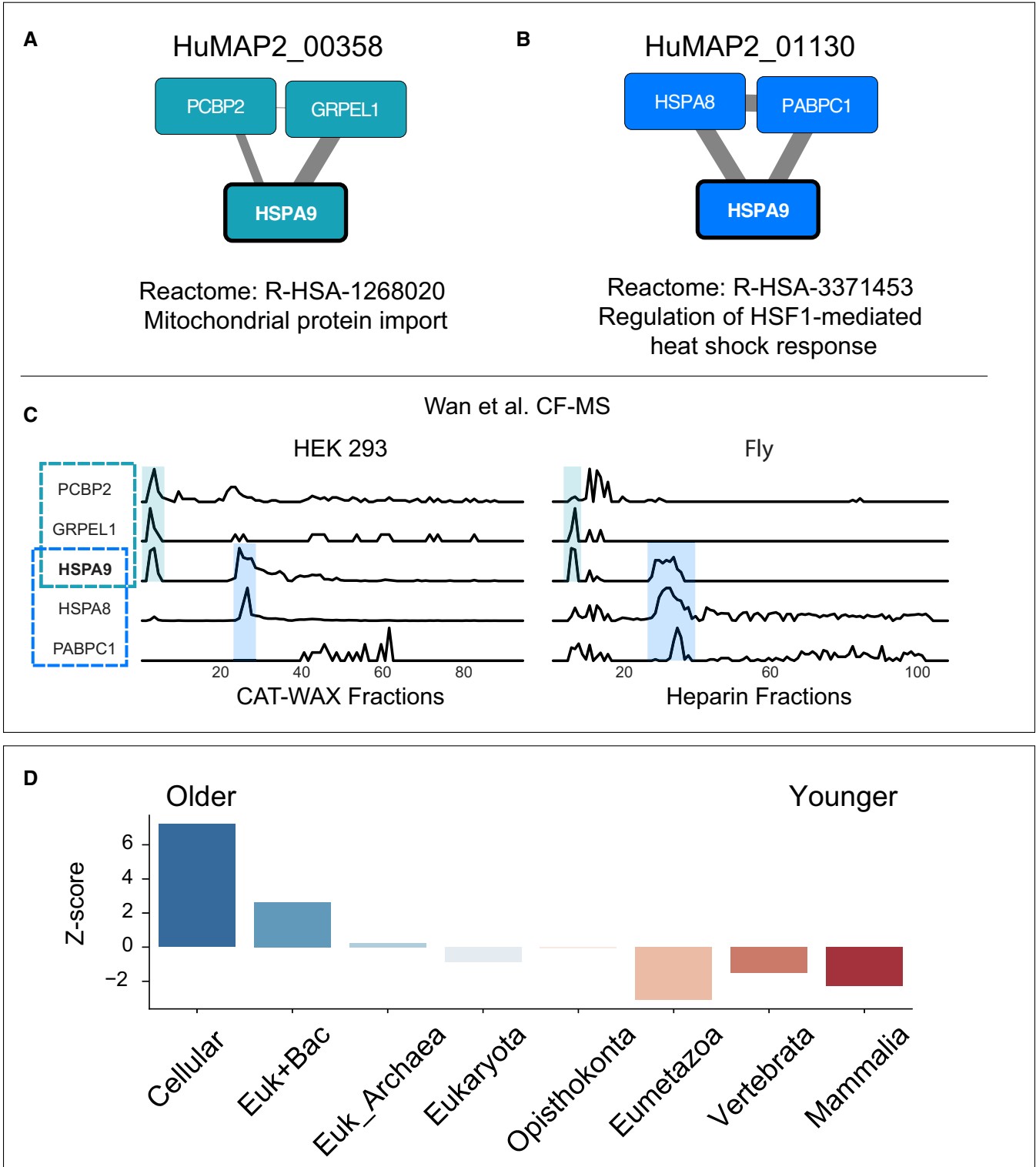

**Figure 3.**

**Figure 3. hu.MAP 2.0 complexes identify promiscuous proteins.**

A, B   Multifunctional protein HSPA9 participates in two distinct complexes, HuMAP2_00358 (A) and HuMAP2_01130 (B). HuMAP2_00358 (turquoise, "high" confidence) is enriched for Reactome annotation "Mitochondrial protein import", a known function of HSPA9. HuMAP2_01130 (blue, "very high" confidence) is enriched for Reactome annotation "Regulation of HSF1-mediated heat shock response", another known function of HSPA9. Weight of network edges represent confidence of interactions.

C   Sparkline elution profiles from two orthogonal biochemical fractionation experiments. HEK 293 cell lysate was separated using a mixed bed ion-exchange column and *Drosophila melanogaster* embryo lysate was separated using a heparin column (Wan *et al*, 2015). HSPA9 elutes in two distinct peaks (shaded) which co-elute with members of the two complexes. *X*-axis represents fraction collected along biochemical separation. *Y*-axis for each row represents observed protein abundance.

D   Promiscuous proteins are older on average than single complex proteins. *Z*-scores for each protein age group were determined by comparing the number of promiscuous proteins to a randomly sampled background set consisting of non-promiscuous proteins (i.e., participating in only one complex).

hypothesize that the promiscuous proteins would be on average older due to younger proteins not having enough evolutionary time to make multiple connections. Figure 3D shows a clear enrichment in older proteins in the set of promiscuous proteins and a depletion for younger proteins (see Materials and Methods). Using gProfiler (Reimand *et al*, 2016) to identify functionally annotations for the older promiscuous proteins, we identify older promiscuous proteins are enriched for Reactome "Metabolism" (adjusted *P*-value ~ $5 \times 10E-8$, g:SCS corrected) among other metabolism-related annotations (Fig EV2). Consistent with this finding, many "moonlighting" proteins are enzymes with multifunctional roles (Jeffery, 2015). Our results suggest a protein's complex membership may play a role in its multifunctional activity.

**Functional annotation of uncharacterized proteins**

High-quality protein complex maps have long been sought after for the purpose of functionally annotating poorly characterized proteins in a genome due to the relationship between physical interaction and biological function (Gavin *et al*, 2002; Ho *et al*, 2002; Wang & Marcotte, 2010). To assess hu.MAP 2.0's ability to functionally annotate uncharacterized proteins, we first tested whether our identified complexes are enriched with literature-curated annotations including Gene Ontology (Ashburner *et al*, 2000), Reactome (Fabregat *et al*, 2016), CORUM (Giurgiu *et al*, 2019), Human Phenotype Ontology (Köhler *et al*, 2014), and KEGG (Kanehisa *et al*, 2014). We see in Fig EV3 that >40% of our complexes are enriched with at least one annotation which is 20.5-fold higher than expected by randomly shuffled complexes. This result shows hu.MAP 2.0 complexes are functionally coherent.

As an example of the utility of hu.MAP 2.0 in annotating poorly characterized proteins, Fig 4A shows two previously unreported

interactions with RNaseH2, CMTR1, and SETD3. The evidence for these interactions is elucidated from co-fractionation mass spectrometry experiments which demonstrate a high degree of correlation over multiple separation column types and multiple organisms, which we have shown suggests a deep conservation in function (Wan *et al*, 2015) (Fig 4B). RNaseH2 is implicated in Aicardi–Goutières syndrome, a monogenic autoinflammatory disorder which mimics *in utero* viral infection of the brain (Reijns & Jackson, 2014). Mechanistically, RNaseH2 degrades RNA fragments of RNA-DNA hybrids including Okazaki fragment RNA primers during DNA replication (Chapados *et al*, 2001). Mutations in RNaseH2 are thought to disrupt the degradation of immuno-stimulating nucleic acids and cause innate immune activation (Reijns & Jackson, 2014). CMTR1 is a mRNA methyltransferase and a known regulator of protein expression of IFN-stimulated genes to restrict viral infection (Williams *et al*, 2020). SETD3, also a methyltransferase, is a human host protein critical for infection of a wide range of viruses and participates in viral replication yet its mode of action is currently unknown (Diep *et al*, 2019). The links we identify between CMTR1, SETD3, and RNaseH2 point to the hypothesis where CMTR1 and SETD3 interact with RNaseH2 to modulate the innate immune response and affect viral replication.

It has been observed that biomedical research is biased toward the study of well-annotated genes and this bias is due less to the physiological importance or disease relevance of the gene but rather to the ease of experimentation using traditional methods (Stoeger *et al*, 2018). Unbiased systematic approaches such as the integration of thousands of mass spectrometry experiments described here provide a powerful tool for closing the gap of uncharacterized proteins. We therefore cross-referenced genes deemed poorly annotated (UniProt (The UniProt Consortium, 2017) annotation score ≤ 3) with hu.MAP 2.0 complexes that were enriched for functional

**Figure 4. Transfer of function annotations to uncharacterized proteins.**

A   SETD3 and CMTR1 are identified as co-complex interactors with the Ribonuclease H2 complex which provides a possible mechanistic explanation for their role in viral infection.

B   Sparkline elution profiles from multiple orthogonal co-fractionation experiments demonstrate a strong degree of co-elution among subunits in the SETD3-CMTR1-RNAse H2 complex. Weight of network edges represents confidence of interactions. *X*-axis represents fraction collected along biochemical separation. *Y*-axis for each row represents observed protein abundance.

C   The uncharacterized protein, C7orf26, is identified as part of the Integrator complex.

D   Sparkline elution profiles show a high degree of correlation between C7orf26 and subunits of the Integrator complex from multiple orthogonal co-fractionation experiments.

E   The association of C7orf26 and Integrator complex is additionally supported by affinity purification mass spectrometry (AP-MS) experiment where C7orf26 is pulled down with Integrator subunit baits.

F   The uncharacterized protein, CCDC9, is identified as co-complex with the exon–exon junction complex (EJC), a ribonucleoprotein complex involved in splicing.

G   Sparkline elution profiles from the independently collected RNA DIF-FRAC size exclusion chromatography (SEC) experiment show CCDC9 co-elutes with known subunits of the EJC when RNA is present (black). The elution profiles also show CCDC9 is sensitive to RNAse A treatment (shift of elution peak between black and red profiles) as are the subunits of the EJC further supporting CCDC9's participation in this known ribonucleoprotein complex.

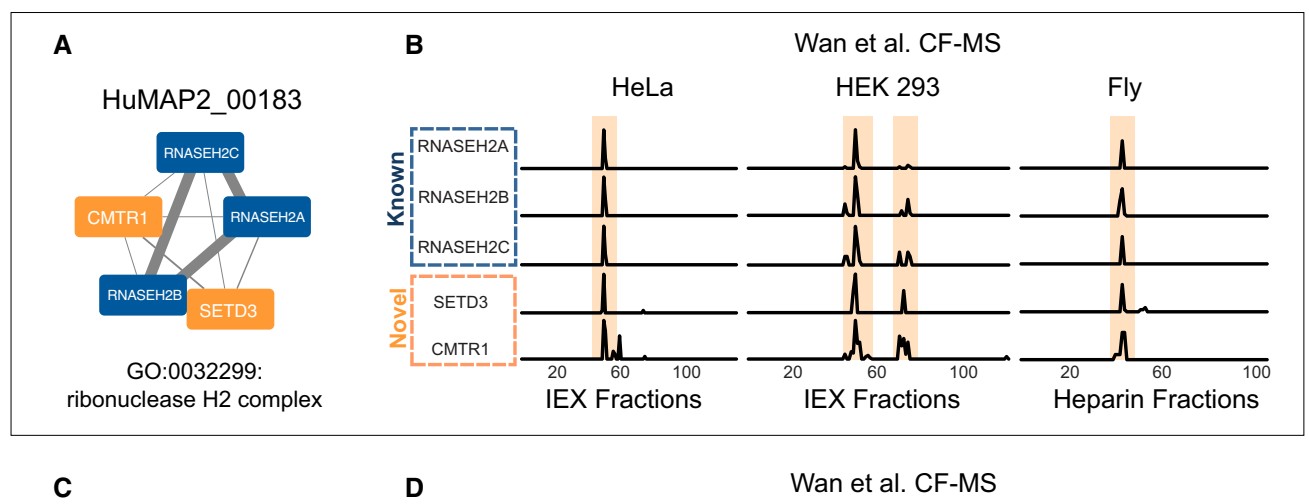

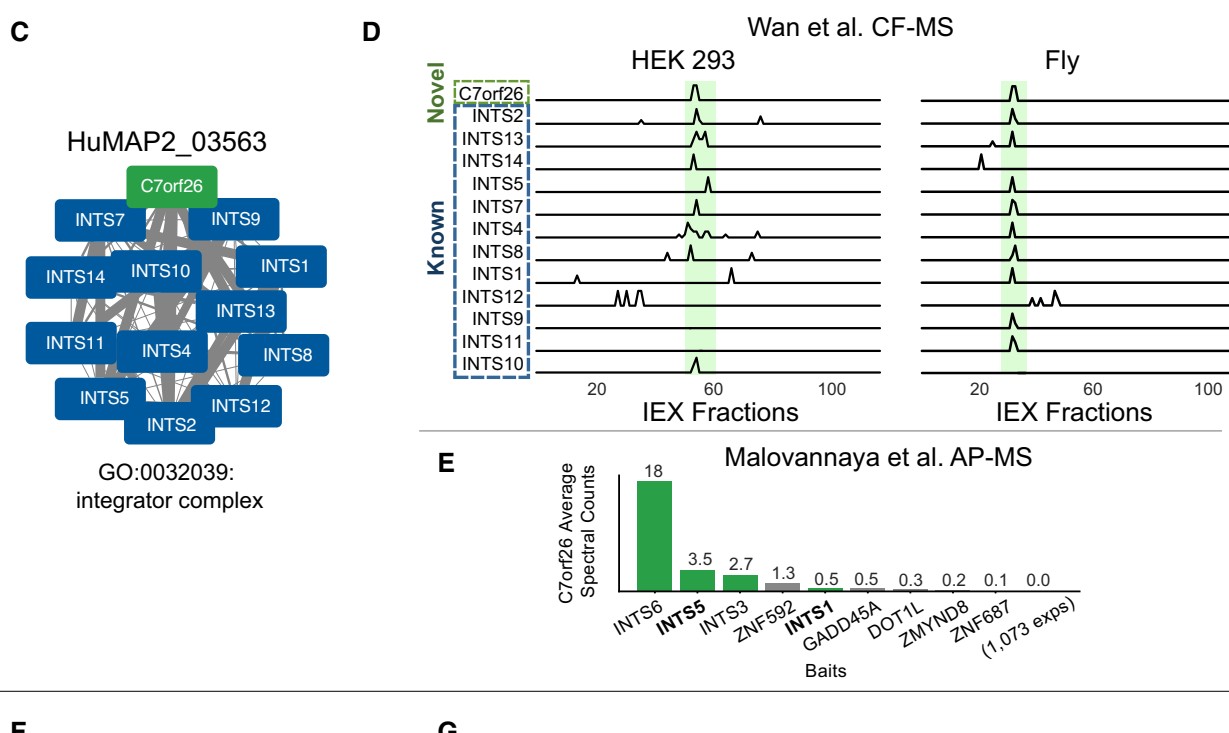

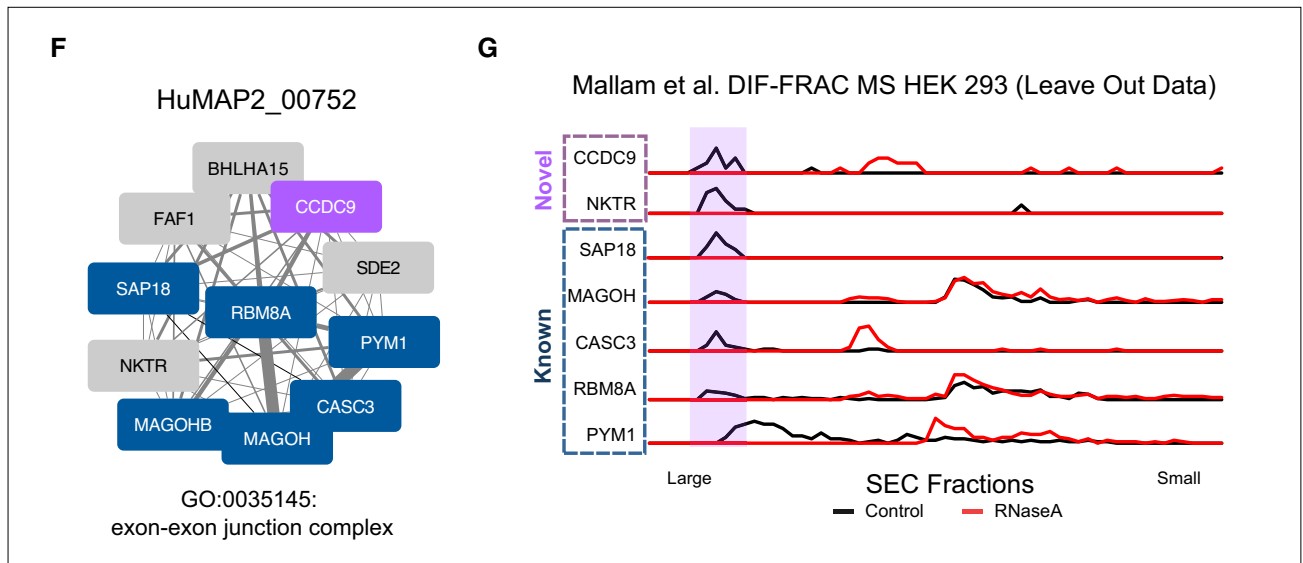

Figure 4.

 Molecular Systems Biology

annotations. We identified 274 proteins in which we could transfer the annotation of the complex to an uncharacterized protein (Dataset EV3). This is an increase of ~ 40% over the same analysis done with hu.MAP 1.0.

Within this set of uncharacterized proteins, we identify C7orf26 as a member of the Integrator complex (Fig 4C). We see strong correlation of C7orf26 with Integrator subunits in co-fractionation experiments from both HEK 293 cells and fly embryos (Fig 4D). In addition, C7orf26 was also identified in several separate affinity purification experiments (Malovannaya *et al*, 2011) where Integrator subunits were the target bait protein (Fig 4E). The consistent results from these orthogonal datasets suggest a role for C7orf26 in the Integrator complex and provide a new function for a completely unannotated protein.

We also observe another uncharacterized protein, CCDC9, as being a member of the exon–exon junction complex (EJC; Fig 4F). The interaction between CCDC9 and the EJC is supported by evidence from Hein *et al* (2015) AP-MS experiments and WMM evidence. Since the EJC is a ribonucleoprotein complex involved in RNA splicing, we searched our previously collected RNA DIF-FRAC mass spectrometry data (Mallam *et al*, 2019) for evidence of CCDC9 as being both a member of the EJC and also associated with RNA. The DIF-FRAC experiment identifies ribonucleoprotein complexes by comparing elution profiles of protein complexes with and without RNAseA treatment. We see in Fig 4G CCDC9 not only co-elutes with EJC subunits but is also sensitive to RNAseA treatment suggesting it is interacting with the EJC while associated with RNA. Importantly, the DIF-FRAC experimental data were not included in the generation of hu.MAP 2.0 and therefore represent an independent assessment of CCDC9's interaction with the EJC. Additionally, CCDC9 was identified as an RNA binding protein in high-throughput screens searching for mRNA-binding proteins (Baltz *et al*, 2012; Castello *et al*, 2012) which is consistent with our observation of CCDC9 participating in a ribonucleoprotein complex.

# Discussion

Herein, we describe the construction of the most accurate and comprehensive protein complex map to date which fills a large gap in our knowledge regarding the composition of functional complexes in the cell. We demonstrate the utility of our map by assigning functions for hundreds of completely uncharacterized proteins, providing testable hypotheses for their characterization. Additionally, we determine the prevalence of proteins that participate in multiple independent protein assemblies including ones with disparate functions suggesting moonlighting functions for the protein. Overall, our results, searchable with a simple web interface (http://humap2.proteincomplexes.org/), establish the utility of hu.MAP 2.0 for furthering our understanding of human protein functions.

### Summary of methodological updates to the hu.MAP pipeline from version 1.0

While the substantial performance gain between hu.MAP 1.0 and hu.MAP 2.0 is largely due to the addition of newly published interactome datasets, we summarize changes between the pipelines here. First, the hu.MAP 1.0 pipeline took advantage of LIBSVM's (Chang & Lin, 2011) grid.py utility to sweep SVM hyper-parameters $C$ and

gamma. The grid.py utility relies on the accuracy metric for evaluation defined as Accuracy = $(TP + TN)/(TP + TN + FP + FN)$ where TP is true positives, TN is true negatives, FP is false positives, and FN is false negatives. Using accuracy as an evaluation metric is problematic in cases of large class imbalance such as protein interaction prediction. Specifically, there are orders of magnitude more negative interactions than positive interactions, and therefore, accuracy can be dominated by the true negatives resulting in a poor predictor of new interactions. In the hu.MAP 2.0 pipeline, we therefore remedied this by developing our own cross-validation utility (https://github.com/marcottelab/protein_complex_maps/tree/master/protein_complex_maps/model_fitting/cross_validation) which uses area under the precision-recall curve (PR-AUC) as a metric to identify optimal $C$ and gamma SVM parameters. PR-AUC does not rely on true negatives for its calculation and is preferred for evaluation of problems with large class imbalance (Davis & Goadrich, 2006).

Second, we simplified the clustering section of the hu.MAP 2.0 pipeline to use only ClusterOne followed by MCL. In the hu.MAP 1.0 pipeline, we observed a gain in performance when we combined results from the Newman clustering method (Newman, 2004) with the MCL results. In the hu.MAP 2.0 pipeline, we do not see such gains and opt for only using MCL.

Third, the hu.MAP 1.0 pipeline did not classify protein complexes by their estimated confidence level. Hu.MAP 2.0 now allows for a quick way for users to evaluate a complex of interest based on a confidence measure. As described above, in the hu.MAP 2.0 pipeline, we use the *k*-clique precision-recall performance measure for evaluating the generated clusterings. The *k*-clique precision-recall measure allows for a finer grain analysis of performance due to the separation of precision and recall terms as opposed to the F-Grand *k*-clique measure used in hu.MAP 1.0. We were then able to select five clusterings that provided a trade-off between precision and recall which ultimately serve as a confidence score for the complexes in those clusterings.

### hu.MAP 2.0 identifies complexes across a broad distribution of biochemical classifications

A primary goal of our work is to identify all stable physical complexes in human cells. While hu.MAP 2.0 substantially increases coverage of stable human complexes, the results ultimately reflect the integrated high-throughput methods' ability to identify these assemblies. For example, high-throughput biochemical purification techniques such as AP-MS and CF-MS are commonly thought to be robust at identifying soluble cytoplasmic protein complexes and less robust at purifying, for example, membrane bound complexes. To determine whether our hu.MAP 2.0 complexes were biased against certain categories of proteins, we used PantherDB (Mi *et al*, 2019) GO analysis to identify enriched and depleted terms. Interestingly, we see GO cellular component terms such as "vesicle membrane" and "mitochondrial membrane" as enriched with 1.27- and 1.41-fold enrichment respectively. Alternatively, we see a depletion in proteins annotated with "plasma membrane" in hu.MAP 2.0 yet only slightly, with a 0.86-fold depletion and a total of over 2,300 proteins with the "plasma membrane" annotation in our complexes. To further investigate the validity of these proteins in our data, we searched, for examples, of plasma membrane complexes. We found several high confidence complexes enriched with "plasma membrane" term including a potassium voltage-gated channel complex (HuMAP2_00584) and a

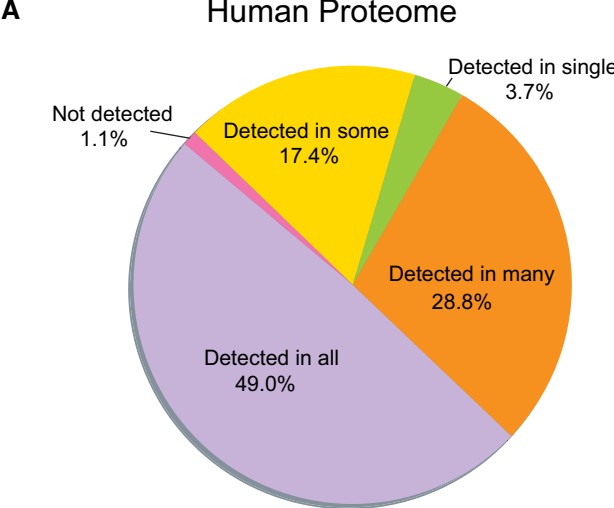

## A  Human Proteome

Not detected
1.1%

Detected in some
17.4%

Detected in single
3.7%

Detected in many
28.8%

Detected in all
49.0%

**Figure 5.  Protein complex map coverage across Human Protein Atlas tissues and cell specificity.**

A  Coverage of all human proteins shows a broad distribution of proteins classified into a range of specificity classes, from detected in all tissues and cells to detected in only a single tissue or cell type.

B  Coverage of hu.MAP 1.0 proteins show a narrower distribution of proteins classified into specificity classes with the majority of proteins detected in many or all tissues and cell types. This suggests hu.MAP 1.0 represented the core cellular machinery.

C  Coverage of hu.MAP 2.0 proteins show a distribution representative of the core cellular machinery shared among all or many tissue and cell types but also shows an increase in cell type specificity with gains in proteins that are only detected in some tissues/cell types.

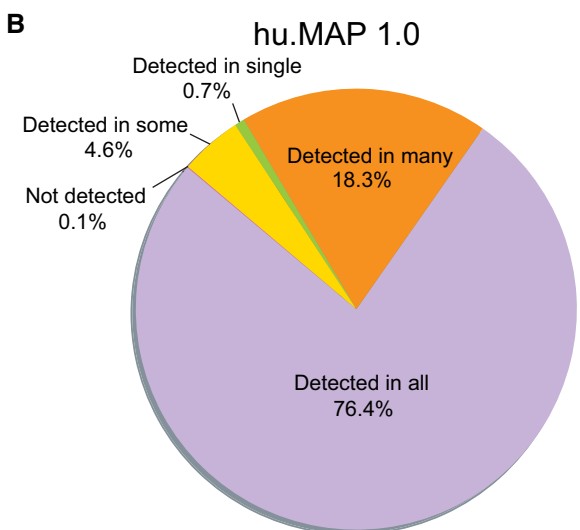

## B  hu.MAP 1.0

Detected in single
0.7%

Detected in some
4.6%

Not detected
0.1%

Detected in many
18.3%

Detected in all
76.4%

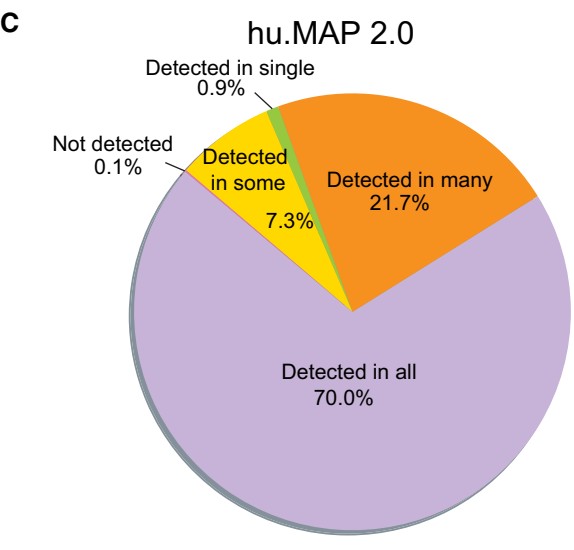

## C  hu.MAP 2.0

Detected in single
0.9%

Not detected
0.1%

Detected in some
7.3%

Detected in many
21.7%

Detected in all
70.0%

fibroblast growth factor receptor complex (HuMAP2_06274). This suggests that while the high-throughput methods we integrate in our pipeline may have some biases against membrane associated proteins, our map still has substantial coverage of these proteins and produces coherent and biologically meaningful complexes.

To further explore biases in our dataset, we considered the degree to which our complexes are distributed across all tissues/cell types or are they cell type specific. To evaluate this potential bias, we considered transcriptomics data in the Human Protein Atlas (across 37 tissues, 43 single cell types, 10 main regions of each mammalian brain, 18 blood cell types, and 69 cell lines) (Uhlén et al, 2015) as an indicator for protein expression. We evaluated these data for the entire human proteome (Fig 5A), proteins identified in hu.MAP 1.0 (Fig 5B), and those identified in hu.MAP 2.0 (Fig 5C). In the entire human proteome, we see a substantial number of proteins likely expressed in all samples (~ 50%) but still many displaying some specificity. In hu.MAP 1.0 and hu.MAP 2.0, we see the vast majority of proteins likely expressed in all samples, ~ 76 and ~ 70%, respectively. This suggests that the complexes we identify are likely to be expressed across many different cellular conditions and cell types. As more and more high-throughput datasets are collected on different cell types and tissues, we expect to identify more cell type-specific complexes upon their integration.

### Completeness of human protein complex map

Our goal in this work is to build a complete and accurate set of protein complexes. We next asked how far have we come in achieving this goal? The size of the human interactome has previously been estimated to contain 154k–369k total interactions (Hart et al, 2006). Here we report 57k distinct interactions equating to roughly 15–37% complete. Consistent with this, if we consider the CORUM benchmark we use for evaluation in Fig 2A as representative of all interactions, we see the majority of the hu.MAP 2.0 precision-recall curve in Fig 2A roughly falls between 15 and 37% recall. Also, making the same assumption for the CORUM benchmark, we can get rough estimates of hu.MAP 2.0's coverage of total protein complexes. The precision and recall metrics used in Fig 2D are robust to any redundancy and therefore give an accurate representation of total coverage. In Fig 2D, we see hu.MAP 2.0 covers > 30% of complexes in the benchmark at a precision of 60%. As described in the section above, many cell type and condition-specific interactions likely make up a large portion of the remaining undiscovered interactions. We expect these interactions to be a focus of future experimentation in order to gain greater coverage of the complete human interactome.

# Materials and Methods

## Reagents and Tools table

| Reagent/Resource | Reference or Source | Identifier or Catalog Number |
|---|---|---|
| **Software** | | |
| MCL Version: 14-137 | https://micans.org/mcl/ (Enright *et al*, 2002) | |
| ClusterOne Version: 1.0 | https://paccanarolab.org/cluster-one/ (Nepusz *et al*, 2012) | |
| LibSVM Version: 321 | https://www.csie.ntu.edu.tw/~cjlin/libsvm/ (Chang & Lin, 2011) | |
| UpSet Version: 0.4.1 | https://upsetplot.readthedocs.io/en/stable/ (Lex *et al*, 2014) | |
| MSBlender Version: ec3b484 | https://github.com/marcottelab/MSblender (Kwon *et al*, 2011) | |
| ProteinComplexMaps Version: 512b9b5 | https://github.com/marcottelab/protein_complex_maps (Drew *et al*, 2017) | |

## Methods and Protocols

### Mass spectrometry dataset collection

Mass spectrometry data and features based on those data used as input into the machine learning classifier were collected from various publications. Specifically, protein interaction features for datasets used in hu.MAP 1.0 (Drew *et al*, 2017), *e.g.*, Wan *et al*, Hein *et al*, Huttlin *et al*, were downloaded from http://hu1.proteincomplexes.org/static/downloads/feature_matrix.txt.gz. Four vector comparison measures were used for co-fractionation data from Wan *et al* including Poisson noise Pearson correlation coefficient, weighted cross-correlation, co-apex score, and MS1 ion intensity distance metric. All four vector comparison measures were applied to each of the 55 fractionation experiments, totaling 220 features. Pairs of proteins were filtered to ensure co-fractionation measures were > 0.5 in at least two species. AP-MS data from (Guruharsha *et al*, 2011) mapped onto human orthologs using InParanoid (Sonnhammer & Östlund, 2015) were represented using the HGSCore value originally downloaded from supplemental table S3 in Guruharsha *et al* AP-MS data from (Malovannaya *et al*, 2011) were represented by the MEMOs (core modules) certainty assignments "approved", "provisional", and "temporary" originally downloaded from supplemental file S1, assigning the scores 10, 3, and 1, respectively. Bioplex 1.0 (Huttlin *et al*, 2015) features were used as originally downloaded from http://wren.hms.harvard.edu/bioplex/data/cdf/150408_CDF_STAR_GRAPH_Ver2594.cdf including NWD Score, *Z* Score, Plate *Z* Score, Entropy, Unique Peptide Bins, Ratio, Total PSMs, Ratio Total PSMs, and Unique:Total Peptide Ratio. For the Hein AP-MS data, the features prey.bait.correlation, valid.values, log10.prey.bait.ratio, and log10.prey.bait.expression.ratio were taken from supplemental table S2 in (Hein *et al*, 2015). The mean value was used across the experiments in the case of multiple entries for a given protein pair. Note, all HumanNet (Lee *et al*, 2011) features were excluded from all model training.

New datasets added for hu.MAP 2.0 were downloaded from original publications or associated dataset web resources as shown in Table 1. The same measures used for Bioplex1.0 were also used for Bioplex2.0 features specifically NWD Score, *Z* Score, Plate *Z* Score, Entropy, Unique Peptide Bins, Ratio, Total PSMs, Ratio Total PSMs, Unique:Total Peptide Ratio and Average Assembled Peptide Spectral Matches. We used the measures from the proximity labeling dataset, Gupta *et al*, for both the ciliated condition and nonciliated

**Table 1. Integrated mass spectrometry datasets.**

| Dataset | Num of experiments | Link |
|---|---|---|
| Wan *et al* (2015) | 5,344 fractions | http://hu1.proteincomplexes.org/static/downloads/feature_matrix.txt.gz |
| Hein *et al* (2015) | 1,125 bait pull-downs | http://hu1.proteincomplexes.org/static/downloads/feature_matrix.txt.gz |
| Bioplex 2.0 (Huttlin *et al*, 2017) (includes Bioplex 1.0) | 5,891 bait pull-downs | https://bioplex.hms.harvard.edu/data/BaitPreyPairs_noFilters_BP2a.tsv |
| Gupta *et al* (2015) | 2 conditions x 58 proximity labeled baits = 116 | http://prohits-web.lunenfeld.ca/ |
| Youn *et al* (2018) | 119 proximity labeled baits | http://www.cell.com/cms/attachment/2118963855/2087347233/mmc2.xlsx |
| Boldt *et al* (2016) | 217 bait pull-downs | https://static-content.springer.com/esm/art%3A10.1038%2Fncomms11491/MediaObjects/41467_2016_BFncomms11491_MOESM835_ESM.xlsx |
| Treiber *et al* (2017) (reprocessed by (Mallam *et al*, 2019)) | 3,004 RNA hairpin pull-down MS runs | https://www.cell.com/cms/10.1016/j.celrep.2019.09.060/attachment/45abb95b-ef3f-4752-8906-dc5eed118480/mmc4.csv |

condition, specifically Average Spectra, Average Saint probability, Max Saint probability, Fold Change, and Bayesian FDR estimate. The same measures were used for the proximity labeling Youn *et al* data but only for the single condition. Measures used for Boldt *et al* data were socioaffinity index ($SA_{ij}$) and individual terms based on the spoke model for where protein $i$ is the bait ($S_{ij}$) and where protein $j$ is the bait ($S_{ji}$). The matrix model term for the socioaffinity index was also used ($M_{ij}$). Mass spectrometry data from (Treiber *et al*, 2017) were downloaded from the Pride web resource (Perez-Riverol *et al*, 2019) (PXD004193) and reprocessed using the MSBlender pipeline (Kwon *et al*, 2011). Full details are described in

(Mallam *et al*, 2019). Only WMM features, described below, were calculated for Treiber *et al*.

HuRI dataset (Luck *et al*, 2020), which was not included in training but was included for evaluation, was downloaded from http://interactome.baderlab.org/data/HuRI.tsv. HuRI protein interactions were ranked based on the number of assays the interaction was identified in.

### Weighted matrix model

To gain additional information on the probability that two proteins interact, we generated additional features using a WMM. The WMM is based on the hypergeometric distribution and is described in (Hart *et al*, 2007) and (Drew *et al*, 2017). Briefly, we used the hypergeometric test (equation (1)), where $k$ represents the number of experiments when both proteins A and B are identified. Variables $n$ and $m$ represent the number of experiments that independently identified protein A and protein B respectively. $N$ represents the total number of experiments. The index $i$ is defined from $k$ to the minimum of $n$ and $m$.

$$p(\#\mathrm{shared\,experiments} \geq k | n, m, N) = \sum_{i=k}^{\min(n,m)} \frac{\binom{n}{i}\binom{N-n}{m-i}}{\binom{N}{m}}. \quad (1)$$

Our implementation of the WMM is based on presence or absence of proteins in individual experiments (e.g., one pull-down). Due to the nature of high-throughput experiments, noise arises in the form of spurious identifications leading to a protein being erroneously called present in the experiment. To deal with this noise, we set arbitrary but sensible cutoffs of the quality of identification required for a protein to be considered present in the experiment. WMM for Bioplex2.0 was calculated only considering experiments for a given protein where the protein had > 2.0 Bioplex2.0 $Z$ score and > 4.0 Bioplex2.0 $Z$ score. WMM based on Gupta *et al* were calculated considering all experiments, > 2 average spectral counts, and > 4 average spectral counts. WMM based on Boldt *et al* were calculated considering all experiments and > 4 spectral counts. WMM based on Treiber *et al* were calculated considering > 2 spectral counts and > 4 spectral counts. WMM based on Youn *et al* were calculated considering all experiments, > 2 spectral counts and > 4 spectral counts. For each calculation, we generate a feature in the form of the negative natural log *P*-value of Equation (1) and the total number of experiments the pair of proteins is observed together (i.e., pair count).

All features, precalculated from original publications and WMM, were combined into 17,564,755 protein pairs × 292 features matrix. Since not all features cover all protein pairs, missing values were filled with 0.0. The final feature matrix can be found in the Data availability section.

### Gold standard test and training set

To create a test and training set of literature-curated protein complexes, we downloaded the complete set of CORUM complexes (Giurgiu *et al*, 2019) version 2017_07_02 (http://mips.helmholtz-muenchen.de/corum/download/corum_2017_07_02.zip) and filtered out all non-human proteins. Complexes were merged to eliminate redundancy, so no two complexes had > 0.6 Jaccard

coefficient. Complexes were then randomly split into test and training sets. A complex was removed from the test or training sets if any pairs of proteins overlapped in the other set. Large complexes > 30 subunits were removed from the test and training complexes. Test and training sets were also generated for pairs of proteins for training the SVM classifier. A pair of proteins was labeled "positive" if both proteins were in the same complex. A pair was labeled "negative" if proteins were in separate complexes. All other pairs were left unlabeled. For test and training pairs, only 10% of pairs from large complexes were considered. Below is the command line used to generate the test and training sets:

./protein_complex_maps/preprocessing_util/complexes/split_complexes.py --input_complexes allComplexes_20170702_geneids_human.txt --random_seed 1234 --size_threshold 30 --subsample_large_complexes 0.1 --remove_large_complexes --remove_largest --merge_threshold 0.6

Additionally, the full test and training sets used in this study can be found in the Data availability section.

### Support vector machine model selection and evaluation

We trained a SVM classifier using Libsvm (Chang & Lin, 2011) to classify pairs of proteins as co-complex protein interactions. We first generated a feature matrix using the features described above where rows are pairs of proteins and columns are features. The feature matrix was further labeled using the gold standard training set described above. We used fivefold cross-validation using only the training set when training to select SVM parameters $C$ and gamma. We evaluated a range of $C$ values (2, 8, 32, 128, 512) and gamma values (0.00048828125, 0.001953125, 0.0078125, 0.03125). As an evaluation metric, we used Area Under the Precision-Recall Curve (AUPRC) averaging across the five cross-validation sets. We identified $C = 512$ and gamma = 001953125 with the highest cross-validated AUPRC. We retrained a full model using all training data using these parameters. We used this final model to predict on all pairs in the feature matrix. The final result is a list of pairs with a corresponding score generated by model.

We evaluated the final model using a precision-recall framework as shown in Fig 2A. We used the scikit-learn python package (preprint: Buitinck *et al*, 2013) to calculate precision and recall for the leave-out gold standard test protein pairs.

For comparisons between datasets as shown in Fig 2A, we generated additional models restricting the features to just those generated from the given dataset keeping the parameters $C$ and gamma fixed. Note, the HuRI dataset was evaluated using the dataset directly as described above.

We additionally evaluated the SVM confidence score for its fidelity to the test set precision value. We observed that the test set precision is consistently higher than the confidence score (Fig EV4). For example, a confidence score as low as 0.02 has ~0.5 precision value.

### Two-stage clustering and parameter set selection

We next used a two-stage clustering approach to identify clusters within the protein interaction network generated by the classification step described above (Fig 2B). First, the network was thresholded based on the SVM score. We then applied the ClusterOne (Nepusz *et al*, 2012) algorithm to identify dense regions in the thresholded network. Further, for each dense region produced by ClusterOne, we applied the MCL (Enright *et al*, 2002) algorithm to

identify clusters. To identify optimal parameters for the score threshold, ClusterOne parameters density and overlap as well as MCL inflation parameter, we generated clusters for various parameter combinations. Specifically, we evaluated a range of parameters: SVM score threshold (1.0, 0.99, 0.97, 0.95, 0.9, 0.8, 0.7, 0.6, 0.5, 0.4, 0.3, 0.2, 0.1, 0.05, 0.01, 0.005, 0.001, 0.0005, 0.0001, 0.00005, 0.00001), ClusterOne max overlap (0.6, 0.7, 0.8), density (0.1, 0.2, 0.3, 0.35, 0.4), and MCL inflation (1.2, 2, 3, 4, 5, 7, 9, 11, 15). We also compared using an unweighted graph as input into ClusterOne versus a weighted graph and observed the unweighted graph had superior performance. A weighted graph was used for the MCL stage clustering. We additionally applied a post-clustering filter that removed nodes from a cluster that lacked edges that scored greater than the SVM score threshold.

To evaluate the clusterings as shown in Fig 2C and D, we used the *k*-cliques method, specifically weighted recall (R_weighted), and weighted precision (P_weighted), which we described previously (Drew *et al*, 2017). Briefly, the *k*-cliques method globally compares a set of clusters to a set of gold standard complexes by comparing cliques derived from the clusters to cliques derived from gold standard complexes. This comparison is done for all clique sizes from size 2 (i.e., pairs) to size *n* (i.e., the size of the largest complex or cluster). A precision and recall value are calculated for all clique sizes. A weighted average is then calculated for both precision and recall across all clique sizes, weighted by the number of clusters with size >= to the clique size. This is to limit the bias effect that larger clusters will have on the final precision or recall value.

We evaluated all clusterings using the *k*-cliques method, comparing to the training set of gold standard complexes. We selected five clusterings that optimize the trade-off between precision and recall as shown in Fig 2C. These five clusterings were then combined into a union set. Table 2 shows the clustering parameters used for the selected clusterings.

We finally evaluated the individual selected clusterings and the union of the selected clusterings using the *k*-clique method by comparing to the leave-out set of gold standard test complexes (Fig 2D). In addition, we compare to previously published complex maps from (Wan *et al*, 2015), Bioplex 1.0 (Huttlin *et al*, 2015), Bioplex 2.0 (Huttlin *et al*, 2017), and our original hu.MAP 1.0 (Drew *et al*, 2017).

### Identification of promiscuous proteins

To identify proteins which participate in multiple complexes, we first determined a set of complexes with limited overlap. To determine the degree with which one complex overlaps another, we developed a 'subcomplex index' (equation (2)) defined as:

$$ SC = \frac{|A \cap B|}{|A|} \tag{2} $$

where *A* and *B* are complexes (i.e., sets of proteins). The 'subcomplex index' is related to the Jaccard index but is normalized by the size of a single complex rather than the size of the union of both complexes as is done in the Jaccard index. We then calculated the subcomplex index for every complex compared to all other complexes. We then generated a set of complexes with limited overlap by selecting complexes that had SC < 0.5 to all other complexes. We then identified all proteins that participated in multiple complexes from this reduced set of complexes.

### Calculation of protein age enrichment for promiscuous proteins

Protein ages were mapped using 'modeAge' in the main_HUMAN.csv file from (Liebeskind *et al*, 2016) *Z*-scores for each age group were determined by comparing the number of promiscuous proteins to a background distribution. The background distribution was calculated by counting the number of randomly sampled non-promiscuous proteins (i.e., proteins that participate in only one complex) in each age group.

### Annotation enrichment, tissue specificity, and overall expression

Annotation enrichment was calculated for GO, Reactome, CORUM, KEGG, and Human Phenotype Ontology (HP) terms using gProfiler (Reimand *et al*, 2016) for each individual complex. All proteins observed in the 15,000 mass spectrometry experiments were used as the background set. Annotations inferred by electronic transfer were ignored.

To evaluate annotation enrichment for all complexes, we first generated a set of shuffled complexes where protein ids were reassigned to new cluster ids. This has the effect of keeping both the number of clusters and the size distribution of clusters the same as the final set of hu.MAP 2.0 complexes. In addition, this also has the effect of keeping the distribution of complexes per protein constant with the final hu.MAP 2.0 complexes as well. Annotation enrichment for the shuffled set of clusters was done as described above. Using this background annotation enrichment from all categories, we calculated a 0.05 false discovery rate threshold.

We used the Human Protein Atlas (HPA) (Uhlén *et al*, 2015) to compare tissue specificity between the full human proteome, hu.MAP 1.0, and hu.MAP 2.0. RNA tissue distribution data were downloaded from: https://www.proteinatlas.org/download/proteinatlas.tsv.zip and mapped to proteins through genenames.

Comparison of overall expression levels between promiscuous and non-promiscuous proteins shown in Fig EV1 was done using HPA using the same proteinatlas.tsv.zip file described above. The median value was calculated for all "Tissue RNA" columns for each

**Table 2. Selected clusterings parameters.**

| Clustering | Confidence | Score threshold | ClusterOne density | ClusterOne overlap | MCL inflation |
|---|---|---|---|---|---|
| 1 | Extremely high | 1.0 | 0.4 | 0.6 | 9 |
| 2 | Very high | 0.7 | 0.4 | 0.6 | 9 |
| 3 | High | 0.5 | 0.4 | 0.7 | 4 |
| 4 | Medium high | 0.04 | 0.4 | 0.7 | 2 |
| 5 | Medium | 0.02 | 0.1 | 0.6 | 2 |

individual promiscuous and non-promiscuous protein, and the resulting distribution was plotted.

## Data availability

hu.MAP 2.0 is available at: http://humap2.proteincomplexes.org/

Protein Complexes List: humap2_complexes http://humap2.proteincomplexes.org/static/downloads/humap2/humap2_complexes_20200809.txt

Protein Interaction Network with probability scores: humap2_ppis_ACC http://humap2.proteincomplexes.org/static/downloads/humap2/humap2_ppis_ACC_20200821.pairsWprob.gz

Cytoscape Network: humap2_protein_complex_map http://humap2.proteincomplexes.org/static/downloads/humap2/humap2_protein_complex_map_20200821.cys

Train Complexes: humap2_train_complexes_ACC http://humap2.proteincomplexes.org/static/downloads/humap2/humap2_train_complexes_ACC_20200818.txt

Test Complexes: humap2_test_complexes_ACC http://humap2.proteincomplexes.org/static/downloads/humap2/humap2_test_complexes_ACC_20200818.txt

Train Positive PPIs: humap2_train_ppis_ACC http://humap2.proteincomplexes.org/static/downloads/humap2/humap2_train_ppis_ACC_20200818.txt

Train Negative PPIs: humap2_neg_train_ppis_ACC http://humap2.proteincomplexes.org/static/downloads/humap2/humap2_neg_train_ppis_ACC_20200818.txt

Test Positive PPIs: humap2_test_ppis_ACC http://humap2.proteincomplexes.org/static/downloads/humap2/humap2_test_ppis_ACC_20200818.txt

Test Negative PPIs: humap2_neg_test_ppis_ACC http://humap2.proteincomplexes.org/static/downloads/humap2/humap2_neg_test_ppis_ACC_20200818.txt

Feature Matrix: humap2_feature_matrix http://humap2.proteincomplexes.org/static/downloads/humap2/humap2_feature_matrix_20200820.featmat.gz

Software pipeline: GitHub https://github.com/marcottelab/protein_complex_maps

Expanded View for this article is available online.

### Acknowledgements

This work was supported by grants from the NIH (01 DK110520, R35 GM122480 to E.M.M.; R01 HL117164 and R01 HD085901 to J.B.W.; K99 HD092613 and L40 HD096554 to K.D.) and the Welch Foundation (F-1515) to E.M.M. The authors acknowledge the Texas Advanced Computing Center (TACC) at The University of Texas at Austin for providing high-performance computing resources that have contributed to the research results reported within this paper. URL: http://www.tacc.utexas.edu.

### Author contributions
KD, JBW, and EMM conceived of the study. KD designed all experiments and analyzed data. KD wrote software including software pipeline and web resources. KD, JBW, and EMM discussed and interpreted the results, and wrote the manuscript.

### Conflict of interest
The authors declare that they have no conflict of interest.

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
