## [Review Process File · Molecular Systems Biology]

hu.MAP 2.0: Integration of over 15,000 proteomic experiments builds a global compendium of human multiprotein assemblies

Kevin Drew, John Wallingford, and Edward Marcotte
DOI: [10.15252/msb.202010016](https://doi.org/10.15252/msb.202010016)

Corresponding author(s): Edward Marcotte (marcotte@icmb.utexas.edu), Kevin Drew (kdrew@utexas.edu)

Review Timeline:

Submission Date:	23rd Sep 20
Editorial Decision:	9th Nov 20
Revision Received:	12th Feb 21
Editorial Decision:	26th Mar 21
Revision Received:	8th Apr 21
Accepted:	9th Apr 21

Editor: Maria Polychronidou

Transaction Report:

Thank you again for submitting your work to Molecular Systems Biology. We have now heard back from the three referees who agreed to evaluate your study. Overall, the reviewers acknowledge the resource value of the study. However, they raise a series of concerns, which we would ask you to address in a major revision.

I think that the recommendations of the referees are rather clear, and therefore I see no need to repeat any of the points listed below. Please let me know in case you would like to discuss in further detail any of the issues raised. All issues raised by the referees would need to be satisfactorily addressed.

On a more editorial level, we would ask you to address the following points.

Reviewer #1:

This manuscript comes from the group published hu.MAP1.0 in Mol. Systems Biology in 2017. Hu.MAP1.0 was an on-line resource presenting a comprehensive set of human multiprotein complexes and protein-protein interactions as determined from the integration of 9000 mass spectrometry experiments designed to interrogate protein complex components on a systems wide scale.

This manuscript describes a new version of the resource, hu.MAP2.0 where an additional 6000 mass spectrometry based experiments have been integrated. The authors have used similar approaches to identify protein complexes, but have created a data resource which is more accurate in terms of the proteins complex components as judged by determining precision and recall. In many ways the manuscript is iterative over the previous publication describing hu.MAP2.0, but the authors have gone some way to unearth new biological insights within their data. The manuscript is mostly well written, but there are elements of the data analysis being used as a black box, with methods referred to in previous papers which are a bit of an archaeological dig to get at important facts. In my opinion this has led to a manuscript which is unclear in parts and lacks essential details, especially about the choice of parameters. I also feel that some of the conclusions are highly speculative without further work. Hu.MAP2.0 however, is an important resource and with some clarifications and softening of some of the conclusions, I think it will be highly suitable for publication in Mol. Systems Biology. In more detail:

1. In the abstract, the authors claim 'we lack a comprehensive set of protein complexes for human cells'. I'm sure human cells have a very comprehensive set of protein complexes, we just don't know what they all are. I suggest this sentence is re-worded.
2. In the abstract I suggest that authors qualify why hu.MAP2.0 is more 'accurate' i.e. more accurate than what?
3. There is no description of how the new data sets were chosen for inclusion in hu.MAP2.0, nor why proximity labelling data was chosen. A short description of what the new data bring to resource should be included.
4. In the methods section the authors say that each individual dataset identifies non-overlapping sets of protein interaction, but this cannot be completely true and many complexes will be sampled by the different methods - I found this statement confusing.
5. The methods section is generally quite superficial. Despite the supplemental material which goes into a little more detail, the section in the main text gives little away and is confusing in parts.
 - a. The statement that WMM balances both the false negative and false positive issues that face both the spoke and matrix models is unclear - Why do they occur in these models and are overcome in the WMM?
 - b. In figure 1A, why does the hu.MAP1.0 data have a bimodal distribution?
6. What are the 292 features the classifier uses that are computed from the mass spectrometry experiments? Are the same features computed irrespective of the sample set? Are there any difference in the structure of the data between AP-MS, CF-MS and proximity labelling data, and if so, how is this handled by any pre-processing step?
7. The authors describe 259 proteins that appear to participate in multi-complexes. Is this backed up by all data sets? If these proteins are very sticky in a cell lysate, could they be erroneously associated with some complexes? Do they in general represent more abundant proteins? Given that they are ancient and metabolic I suspect the answer is yes, and if this is the case is their apparent involvement in multiple complexes be down to technical limitations of the methods used?
8. Do the 259 'moonlighting' proteins appear in the DIF-FRAC data that is interrogated later in the manuscript? Many metabolic enzymes bind RNA and hence it would be useful to know whether RNA binding is part of a 'moonlighting' function?
9. The conclusion that CMTR1, SETD2 and RNaseH2 are involved in the modulation of the innate

immune response and viral replication is attractive but only conjecture. This conclusion should be softened as it needs empirical testing.

10. Had the authors considered comparing their data with that of the recent Nat. Biotech paper from the Rappsilber groups (doi: 10.1038/s41587-019-0298-5), that charts co-regulation of human proteins by machine learning approaches as applied to a great many datasets. Although these data are not limited to physically interacting proteins, it would be interesting to see if co-regulation of complex components is prevalent in the data within hu.MAP2.0.

11. I struggle with figure 2B - why were the five clustering chosen, what were the 1700 clustering parameters - more detail is needed.

12. In figure 4C why are SDE1 and FAF1 part of the complex - there is no evidence from the spark-line trace?

13. Why was the Treiber et al data re-processed using MSBlender?

14. In the WMM section what was the rationale behind using different cut-offs for the different datasets?

In general:

1. I think it would be good if the authors give an overall assessment of the hu.MAP2.0 data. For example how biased are the data? Are there certain categories of protein that are under-represented? What types of protein complex have been added by the inclusion of proximity labelling data? I would imagine that transient interactors would fare better in proximity labelling data, but this would mean little overlap with AP-MS and CF-MS data. How was this handled?

2. Could the authors summarise what was different about the analytical pipeline used in this manuscript compared with what was applied in the creation of hu.MAP1.0

Reviewer #2:

Summary

The manuscript submitted by Drew et al with the title "hu.MAP 2.0: Integration of over 15,000 proteomic experiments builds a global compendium of human multiprotein assemblies" describes an updated version of the human protein complex dataset hu.MAP 1.0. Various human proteome-scale MS-based protein interaction resources have been published over the last years. By integrating the raw data of several of these resources into a machine learning approach, a more complete and accurate set of human protein complexes termed hu.MAP 1.0 had been generated by these authors in the past. By adding more recently published MS datasets (an increase from 9000 to 15000 MS experiments), the authors generated hu.MAP 2.0, which doubles the number of identified complexes, increases the number of proteins that are part of this complex by about 20% (2000 proteins), however, interestingly does not lead to a notable increase in the number of reported protein interactions. The authors explore hu.MAP 2.0 by identifying likely pleiotropic proteins (those that have multiple different functions) and by predicting functions to uncharacterized proteins based on their membership in protein complexes with enriched functions. Based on this study, the authors conclude that hu.MAP 2.0 is more complete and more accurate compared to its previous version. The methodology used to generate hu.MAP 2.0, to the best of my understanding, is identical to the approach used to generate hu.MAP 1.0. The data is downloadable and searchable at a dedicated web server. This study and web server would be of use to biologists to look up information for their favourite protein and for systems biologists who seek human protein interaction data for their integrative analyses.

General remarks

The increase in accuracy of hu.MAP 2.0 compared to 1.0 is less than 10% for recall and precision. The two analyses performed with hu.MAP 2.0 (identification of pleiotropic proteins and function prediction for uncharacterized proteins) are not compared in terms of their output to hu.MAP 1.0. It is thus difficult to judge the advances of hu.MAP 2.0 beyond the slight increase in accuracy and higher coverage. To more convincingly show the advances of hu.MAP 2.0 over 1.0, the authors would need to compare hu.MAP 1.0 with 2.0 in more ways as currently done in the manuscript.

Major points

The differences in coverage as summarized above are not described in the manuscript (as it seems). Putting these or similar numbers in would help the reader judge the advances of hu.MAP 2.0.

The first sentence of the Results and Discussion section makes the following statement: "we can ask questions that were previously hindered by less accurate maps". To support this statement, the authors would need to show how the outcome of analyses improved using hu.MAP 2.0 compared to 1.0. For example, what is the increase in number of uncharacterized proteins for which hu.MAP 2.0 was able to make a function prediction compared to using hu.MAP 1.0? The same comparison could be done for the number of pleiotropic proteins identified.

The validity of predicted pleiotropic proteins is evidenced by discussion of a single example, HSPA9. More systematic analyses seem more appropriate to validate the predictions as a whole. For example, the authors could try to show that the number of non-redundant GO terms for these proteins is higher than for the non-pleiotropic proteins that are also part of the identified protein complexes. The authors state the described example HSPA9 "demonstrates the ability of our complex map to identify multifunctional promiscuous proteins". This seems overstated without any additional systematic validations of the predictions.

The authors identify that proteins that are part of multiple protein complexes in their dataset tend to be older/more conserved. I think in the HuRI paper, the authors showed that features like age and essentiality of genes correlate with their expression levels. Because MS has a well-known bias towards more highly expressed genes, the question arises whether the correlation between age and number of complex membership is confounded by an expression bias in the underlying MS-based datasets. Younger proteins might be less often detected in protein complexes because they are more lowly expressed and thus more often missed by MS. To support the observed trend between age and protein complex membership, it could be tested whether there is no significant trend between protein complex memberships and expression levels or at least, if such a trend was much weaker than the one observed between age and complex membership.

Does the performance assessment shown in Fig 2a assume that all interactions in a dataset that are not in CORUM are false positives? This is generally problematic but especially problematic for the HuRI dataset, which was generated using a very orthogonal method (Y2H) compared to MS-based techniques used in this manuscript. Y2H cannot find indirect associations, which dominate co-complex data (as evidenced in the HuRI paper), and more transient interactions that might be more easily detected by Y2H are usually lost in MS-based experiments due to washing procedures. While it is important to continuously remind readers about the technical and content-wise differences between Y2H-based and MS-based datasets, this analysis requires a more detailed interpretation compared to what is currently provided in the manuscript to avoid mis-interpretation by readers.

In the section entitled "Annotation enrichment" the authors describe the randomization of the protein complexes to calculate enrichments. From the brief description it remains unclear whether the number of complexes in which a protein participates was kept constant. This is key as proteins with a higher number of complex memberships are likely to be also better functionally annotated.

The results page on the hu.MAP 2.0 web portal would benefit from providing information about the source datasets in which reported associations were identified. This would significantly help biologists to plan their experiments for the follow-up of identified interesting protein associations.

Minor points

Individual figure panels should have individual letters and not 3 panels for example assigned to Figure 4b.

Figure 1b is referenced in the text to show that the WMM "provides evidence of interactions between many pairs of proteins not covered in the other datasets". How is this shown in this figure? This is an interesting point. It would be worth clarifying this simply by adding the count of unique interactions to the manuscript.

The hu.MAP 2.0 web portal provides various datasets for download that are not provided as supplementary information in the manuscript. It would be helpful to the reader to point to the web portal in the manuscript where mentioned datasets are available for download, i.e. the 292 features.

CORUM asks users on their website to cite a more recent publication than the one cited in this manuscript, which seems appropriate given the more recent version of CORUM that has been used in this manuscript.

The two if not the last 3 sections of the methods part might be better placed in the results part as they describe results of this study as also evidenced by the figures cited in these sections. The supplementary methods section contains a lot of information that is missing in the very general and sometimes uninformative description of the methods section. Moving some of this information from the supplement to the actual methods section would facilitate understanding of the methods by the reader.

To describe proteins with multiple different functions, the term pleiotropy is probably more commonly used than promiscuity that is used in this manuscript.

The authors hypothesize that "promiscuous proteins would be on average older due to younger proteins not having enough evolutionary time to make multiple connections". Is there any previous work that would support this idea and which could be cited?

All examples picked to illustrate findings have evidence from Wan et al. It remains unclear whether the selected protein associations were also found in any of the other datasets? If yes, it might be worth adding this information to the manuscript text.

The description of the WMM in the methods section is somewhat confusing and actually much clearer in Hart et al. It might be helpful to explain what the authors mean by "experiment", i.e. one MS run of a pulldown or eluted fraction?

In equation 1, the index i is used. Where does i start? At 1?

The authors describe that equation 1 has been calculated with different cutoffs for the different source datasets. Why?

Reviewer #3:

The article "hu.MAP 2.0: Integration of over 15,000 proteomic experiments builds a global compendium of human multiprotein assemblies" by Drew et al presents an update on the hu.Map project, which the authors originally published in a 2017 MSB paper. The authors use their established computational pipeline from that paper to predict human protein complexes, by integrating various types of proteomics data from different sources. Features from the original articles are collected in a feature matrix together with a set of new features obtained by the authors' own WMM procedure, a SVM machine learning classifier is used to identify high-quality binary protein - protein interactions and a clustering algorithm identifies protein complexes. The resulting hu.Map 2.0 set of protein complexes appears to be the largest, and most accurate, set of computationally compiled protein complexes to date, and therefore presents an very valuable resource to the biomedical community.

To be clear, in contrast to the original hu.Map paper, this manuscript does not present anything new in terms of methods or approach, as far as I can tell. What is new here is that more datasets have been added and processed by the computational pipeline established by the authors. However, this significantly increased both accuracy and coverage of human protein complexes and therefore would be, in my opinion, a very useful publication from a resource perspective. For context, the authors' evaluation convincingly shows that huMap 2.0 clearly outperforms related resources, such as BioPlex (Huttlin et al, Cell 2015 and Nature 2017), which are in fact incorporated in huMap.

The pipeline itself is impressive and has been previously reviewed and validated. Therefore, I only have a few minor points that I think the authors should address:

- I find the title strongly misleading. The reference to 15,000 proteomics experiments sounds like the authors re-processed the MS raw files from scratch, as for example is done by the proteomicsDB project from the Kuester lab. This impression is reinforced by referring to input data as "raw features" and more or less explicitly stated in the sentence "To construct hu.MAP 2.0 we integrated over 15,000 previously published mass spectrometry experiments using our custom machine learning framework". However, here the authors download the processed tables and scored pairwise interactions from individual publications and use those as input for the machine learning, not the actual "raw" proteomics experiments. It is also not clear where the number 15,000 comes from (I assume this is the sum of MS raw files used by the combined set of input datasets). In any case, this is not just an issue of semantics. While re-processing raw files from scratch may not be necessary, or even possible, for a dataset of this size, without doing so one cannot fully integrate the data either. This becomes an issue when analysing interactions between protein isoforms, for example when original articles mapped the same interaction to different isoforms because of the precise set of peptides observed in each experiment.

- For the PR curve, how does it look when you use just the WMM features?

- Related to this, why is the performance of the HuRi data set so poor? Yeast 2 hybrid screens

have been notorious for their high false positive rates in the past, but this striking apparent lack of performance could suggest that something else is going on. Could this reflect the fact that most of the interactions presented by huMap are in fact indirect interactions?

- On the subject of direct vs indirect interactions, can you estimate the relative fraction of each? I imagine that the huMap pipeline enriches for indirect interactions, due to the nature of the input data and using CORUM as a training set.

- The biological follow-up and validation aspect is quite poor. Although the authors claim that the approach can be used to functionally characterise unknown proteins and identify the multiple functions of moonlighting proteins, there is no experimental demonstration of this claim, apart from highlighting a few previously known examples.

- The statement "Unfortunately, we still lack a comprehensive set of protein complexes for the human cell" (page 2). Is there any evidence to support this statement, or are there any estimates about how many complexes actually exist? Or perhaps more relevantly, could one estimate when this approach will reach saturation? 7,000 "protein assemblies" seems to be a very large number already.

- In the methods section, "Each individual experimental dataset identifies nonoverlapping sets of protein interactions" is misleading. They are hopefully partially overlapping.

- The "upset" plot is very useful and clearly better than a set of Venn Diagrams. However, it would be good to describe how it works in the legend, or provide a reference.

We first want to say we greatly appreciate the time and effort the reviewers put into reading our manuscript. We believe the suggestions provided have given us the opportunity to substantially strengthen our manuscript. We address each of the reviewers specific points below.

Reviewer #1:

This manuscript comes from the group published hu.MAP1.0 in Mol. Systems Biology in 2017. Hu.MAP1.0 was an on-line resource presenting a comprehensive set of human multiprotein complexes and protein-protein interactions as determined from the integration of 9000 mass spectrometry experiments designed to interrogate protein complex components on a systems wide scale.

This manuscript describes a new version of the resource, hu.MAP2.0 where an additional 6000 mass spectrometry based experiments have been integrated. The authors have used similar approaches to identify protein complexes, but have created a data resource which is more accurate in terms of the proteins complex components as judged by determining precision and recall.

In many ways the manuscript is iterative over the previous publication describing hu.MAP2.0, but the authors have gone some way to unearth new biological insights within their data.

The manuscript is mostly well written, but there are elements of the data analysis being used as a black box, with methods referred to in previous papers which are a bit of an archaeological dig to get at important facts. In my opinion this has led to a manuscript which is unclear in parts and lack essential details, especially about the choice of parameters.

I also feel that some of the conclusions are highly speculative without further work.

Hu.MAP2.0 however, is an important resource and with some clarifications and softening of some of the conclusions, I think it will be highly suitable for publication in Mol. Systems Biology.

In more detail:

1. In the abstract, the authors claims 'we lack a comprehensive set of protein complexes for human cells'. I'm sure human cells have a very comprehensive set of protein complexes, we just don't know what they all are. I suggest this sentence is re-worded.

We agree with the reviewer that this sentence was confusing and have updated in the manuscript to be:

“Unfortunately, we lack knowledge of the comprehensive set of identities of protein complexes in human cells.”

2. In the abstract I suggest that authors qualify why hu.MAP2.0 is more 'accurate' i.e. more accurate than what?

We have updated the abstract to include a more specific statement of our comparison:

“We show our resource, hu.MAP 2.0, is more accurate and comprehensive than previous state of the art high throughput protein complex resources ...”

3. There is no description of how the new data sets were chosen for inclusion in hu.MAP2.0, nor why proximity labelling data was chosen. A short description of what the new data bring to resource should be included.

We agree that the manuscript would gain from a short discussion on this topic and we have added the text below:

“Our rationale for including these datasets were two fold. First, each dataset samples a different set of bait proteins which provides increased coverage of the interactome. Second, the methods are orthogonal and complementary, where affinity purification targets stable interactions, and the proximity labeling datasets potentially also capture transient *in vivo* interactions.”

4. In the methods section the authors say that each individual dataset identifies non-overlapping sets of protein interaction, but this cannot be completely true and many complexes will be sampled by the different methods - I found this statement confusing.

We agree with the reviewer that this was confusing and appreciate this being drawn to our attention. We have updated the manuscript to now read:

“Each individual experimental dataset identifies different sets of protein interactions and therefore combining them results in a more accurate and comprehensive set of interactions.”

5. The methods section is generally quite superficial. Despite the supplemental material which goes into a little more detail, the section in the main text gives little away and is confusing in parts.

We now incorporated the supplemental methods section directly within the main text.

a. The statement that WMM balances both the false negative and false positive issues that face both the spoke and matrix models is unclear - Why do they occur in these models and are overcome in the WMM?

We now expand on this point in the text:

“The WMM balances both the false negative and false positive issues that face both the spoke and matrix models and therefore is capable of identifying novel interactions. More specifically, since a spoke model only considers interactions between a bait protein and a prey protein, all true interactions between prey proteins are missed leading to high false negative rates for the spoke model. Alternatively, a naive matrix model does consider interactions between prey proteins limiting false negatives but does so by treating all prey pairs equally. Some of these prey pairs will participate in the same complex but since proteins participate in multiple complexes, two prey proteins pulled down by the same bait are not guaranteed to interact. This leads to a high degree of false positives for the naive matrix model. The WMM considers all prey pairs as interactors but weights them according to the frequency they occur together while

controlling for “frequent flyer” or “sticky” proteins. Therefore, by considering all prey pairs the WMM has better false negative rates than the spoke model, and by accurately measuring the specificity of the prey pairs the WMM has better false positive rates than the naive matrix model.”

b. In figure 1A, why does the hu.MAP1.0 data have a bimodal distribution?

The bimodal distribution in the hu.MAP1.0 curve in figure 2A is likely due to a change from predictions with multiple lines of evidence to predictions with a single dominant line of evidence. In particular, if we look at predictions in the “shoulder” (roughly between recall of 0.2 and 0.4) we see an increase in Bioplex1 WMM score but lower Bioplex1 spoke model scores compared to predictions made immediately before the shoulder. We do note that our previous publication describing hu.MAP1 shows a precision recall curve without a shoulder. The differences between the models is likely due to the model for the hu.MAP 1.0 in our current manuscript prioritized predictions with multiple lines of evidence over a high scoring single piece of evidence.

6. What are the 292 features the classifier uses that are computed from the mass spectrometry experiments? Are the same features computed irrespective of the sample set? Are there any difference in the structure of the data between AP-MS, CF-MS and proximity labelling data, and if so, how is this handled by any pre-processing step?

We thank the reviewer for this request and we apologize for the oversight of not including the complete list of features in the original manuscript. We now list all features in the method section. Further, we updated the text to include how we handle missing data in our feature matrix which is to add in zero values. Additionally, we point the readers to the url in which the full feature matrix can be downloaded from our website.

7. The authors describe 259 proteins that appear to participate in multi-complexes. Is this backed up by all data sets? If these proteins are very sticky in a cell lysate, could they be erroneously associated with some complexes? Do they in general represent more abundant proteins? Given that they are ancient and metabolic I suspect the answer is yes, and if this is the case is their apparent involvement in multiple complexes be down to technical limitations of the methods used?

The complexes that encompass promiscuous proteins are not expected to be backed by all datasets due to the fact that datasets are not completely overlapping in terms of their coverage.

Since “sticky” proteins are down weighted at the feature (e.g. weighted matrix model) and machine learning levels we also do not believe this to be a problem. Consistent with this, we observe the majority of promiscuous proteins participate in only two complexes and the highest number of complexes a protein participates in is four.

In regards to an abundance difference between promiscuous and non-promiscuous proteins, we compared expression levels from Human Protein Atlas between the groups and see a substantial overlap between the two sets suggesting that abundance does not explain promiscuous proteins. See newly added Extended Figure 1.

8. Do the 259 'moonlighting' proteins appear in the DIF-FRAC data that is interrogated later in the manuscript? Many metabolic enzymes bind RNA and hence it would be useful to know whether RNA binding is part of a 'moonlighting' function?

This is an interesting question as one could imagine promiscuous proteins as RNA modules used repeatedly for regulation of multiple complexes or several other hypotheses along these lines. When we cross reference the 259 promiscuous proteins with the 1012 RNA associated proteins identified in our DIFFRAC data, we see an overlap of only 24. While this is a significant overlap as determined by the hypergeometric test (p -value = 0.003), it does not seem to be the overall principle driving promiscuous proteins. That being said, not all RNA binding proteins have been identified, limiting this analysis, and therefore this idea may be an interesting avenue to evaluate again in the future.

9. The conclusion that CMTR1, SETD2 and RNaseH2 are involved in the modulation of the innate immune response and viral replication is attractive but only conjecture. This conclusion should be softened as it needs empirical testing.

We have softened the language regarding this conclusion and now reads:

“The links we identify between CMTR1, SETD3, and RNaseH2 point to the hypothesis where CMTR1 and SETD3 interact with RNaseH2 to modulate the innate immune response and affect viral replication.”

10. Had the authors considered comparing their data with that of the recent Nat. Biotech paper from the Rappasilber groups (doi: 10.1038/s41587-019-0298-5), that charts co-regulation of human proteins by machine learning approaches as applied to a great many datasets. Although these data are not limited to physically interacting proteins, it would be interesting to see if co-regulation of complex components is prevalent in the data within hu.MAP2.0.

We thank the reviewer for the suggestion and now include a comparison to the Kustatscher et al. dataset which shows a highly significant overlap among the datasets.

“Recently, a co-regulation map based on protein expression was shown to capture relationships among proteins that do not necessarily interact or co-localize²⁵. This dataset therefore provides an independent test of the quality of our protein interactions. When we compared the highest confidence hu.MAP 2.0 interactions to the most co-expressing pairs in Kustatscher et al., we see a highly significant overlap (p -value < $10E-10$) indicating a high degree of consistency between the orthogonal datasets. “

11. I struggle with figure 2B - why were the five clustering chosen, what were the 1700 clustering parameters - more detail is needed.

We thank the reviewer for this comment. We have now added a panel to figure 2 which shows our clustering workflow which we hope addresses any confusion. We have also edited the main text to clarify the discussion on the clustering as it helps to rank the confidence in the identified protein complexes.

12. In figure 4C why are SDE1 and FAF1 part of the complex - there is no evidence from the spark-line trace?

In figure 4C, we report SDE2 and FAF1 as part of the complex from evidence integrated into our machine learning pipeline. Specifically, the Hein et al. AP-MS data provides evidence for SDE2 as a member of the complex through its interaction RBM8A and the Wan et al. fractionation data provides evidence for an interaction between FAF1 and both MAGOHB and MAGOH. The spark-line plot shown in figure 4G is leave out data that was not included in the machine learning pipeline and therefore is a true independent test of the prediction. As the reviewer notes, SDE2 and FAF1 do not co-elute with the rest of the complex in the independent test. There are a number of reasons why this may be the case including condition or cell type specific interactions, unstable interactions in experimental conditions, or false positives. We agree with the reviewer this is confusing and have removed the spark-lines for SDE2 and FAF1 to focus the exposition specifically on CCDC9.

13. Why was the Treiber et al data re-processed using MSBlender?

The supplemental tables of Treiber et al. were incompatible with calculation of the weighted matrix model. In particular, the published tables report spectrum counts for ambiguous protein groups which would need to be collapsed. We used the MSBlender pipeline with a non-redundant proteome for this purpose.

14. In the WMM section what was the rationale behind using different cut-offs for the different datasets?

Thank you for this comment. We now clarify the use of cut-offs with the included text:

“Our implementation of the WMM is based on presence or absence of proteins in individual experiments. Due to the nature of high throughput experiments, noise arises in the form of spurious identifications leading to a protein being erroneously called present in the experiment. To deal with this noise, we set arbitrary but sensible cutoffs of the quality of identification required for a protein to be considered present in the experiment.”

In general:

1. I think it would be good if the authors give an overall assessment of the hu.MAP2.0 data. For example how biased are the data? Are there certain categories of protein that are

under-represented? What types of protein complex have been added by the inclusion of proximity labelling data? I would imagine that transient interactors would fare better in proximity labelling data, but this would mean little overlap with AP-MS and CF-MS data. How was this handled?

We thank the reviewer for this comment and agree this is a useful discussion to include in our manuscript. We have therefore written a section titled "**hu.MAP 2.0 identifies complexes across a broad distribution of biochemical classifications**" describing the biases in our data. One particular bias we explore is the potential under-representation of membrane associated proteins in our dataset. Interestingly, we observe an enrichment for mitochondrial membrane proteins as well as vesicle membrane proteins. On the other hand, we observe a slight depletion in plasma membrane associated proteins yet we still uncover thousands of proteins annotated as plasma membrane associated in our complexes. We highlight several membrane specific complexes including potassium channels and growth factor receptors as a demonstration of our ability to capture this class of proteins. Further we discuss the specificity of proteins we identify across different tissues and cell types (added Figure 5) in which we observe most identified complexes are likely seen in many if not all cell types.

In regards to the comment on the proximity labeling data, this is a very good point as proximity labeling techniques have the potential to identify transient interactors as the reviewer said. That being said, we do not believe the proximity labeling data is allowing us to capture many new transient interactions likely due to there are far fewer proximity labeling experiments included (>200) as compared to AP-MS (>7,000). As a result of this, there are only four complexes all of which are binary that have evidence only from proximity labeling, and no other sources. We do however see many complexes with a mixture of evidences including AP-MS, proximity labeling and co-fractionation, telling us that proximity labeling is providing a supporting role for the identification of complexes. We believe as proximity labeling methods become more high throughput and those data become available, we will begin to capture more of the transient interactors in our database.

2. Could the authors summarise what was different about the analytical pipeline used in this manuscript compared with what was applied in the creation of hu.MAP1.0

This is a very useful suggestion. We now include a section that highlights differences between the hu.MAP 1.0 and hu.MAP 2.0 pipelines subtitled: "**Summary of methodological updates to the hu.MAP pipeline from version 1.0**".

Reviewer #2:

Summary

The manuscript submitted by Drew et al with the title "hu.MAP 2.0: Integration of over 15,000 proteomic experiments builds a global compendium of human multiprotein assemblies" describes an updated version of the human protein complex dataset hu.MAP 1.0. Various

human proteome-scale MS-based protein interaction resources have been published over the last years. By integrating the raw data of several of these resources into a machine learning approach, a more complete and accurate set of human protein complexes termed hu.MAP 1.0 had been generated by these authors in the past. By adding more recently published MS datasets (an increase from 9000 to 15000 MS experiments), the authors generated hu.MAP 2.0, which doubles the number of identified complexes, increases the number of proteins that are part of this complex by about 20% (2000 proteins), however, interestingly does not lead to a notable increase in the number of reported protein interactions. The authors explore hu.MAP 2.0 by identifying likely pleiotropic proteins (those that have multiple different functions) and by predicting functions to uncharacterized proteins based on their membership in protein complexes with enriched functions. Based on this study, the authors conclude that hu.MAP 2.0 is more complete and more accurate compared to its previous version. The methodology used to generate hu.MAP 2.0, to the best of my understanding, is identical to the approach used to generate hu.MAP 1.0. The data is downloadable and searchable at a dedicated web server. This study and web server would be of use to biologists to look up information for their favourite protein and for systems biologists who seek human protein interaction data for their integrative analyses.

General remarks

The increase in accuracy of hu.MAP 2.0 compared to 1.0 is less than 10% for recall and precision. The two analyses performed with hu.MAP 2.0 (identification of pleiotropic proteins and function prediction for uncharacterized proteins) are not compared in terms of their output to hu.MAP 1.0. It is thus difficult to judge the advances of hu.MAP 2.0 beyond the slight increase in accuracy and higher coverage. To more convincingly show the advances of hu.MAP 2.0 over 1.0, the authors would need to compare hu.MAP 1.0 with 2.0 in more ways as currently done in the manuscript.

We understand the reviewer's concern and now include additional comparisons between hu.MAP 2.0 and hu.MAP 1.0. We first want to gently push back on the idea that the increase in performance of hu.MAP 2.0 over hu.MAP 1.0 is marginal. When the reviewer mentions 10% improvement of hu.MAP 2.0 over 1.0, we assume the reviewer is referring to absolute recall as defined on the x-axis of Figure 2A. If we look at the absolute recall of both hu.MAP 1.0 and hu.MAP 2.0 at a defined precision of .5, we see a recall of 0.18 and 0.3 respectively. This results in an increase of 12% absolute recall of hu.MAP 2.0 over hu.MAP 1.0. While we believe this to be a large boost in performance, it is even more substantial when viewed in the light that all current state of the art datasets have low recall. When we take this into account and measure the increase in recall relative to hu.MAP 1.0, we see that hu.MAP 2.0 increases recall by 67%. We also evaluated the increase of total interactions identified at a very high confidence (defined precision of 0.9) which resulted in an increase of >1,880 interactions reported in hu.MAP 2.0 over hu.MAP 1.0.

In regards to the improvement of hu.MAP 2.0 over hu.MAP 1.0 in terms of the set of promiscuous proteins and function annotation of uncharacterized proteins, we now include in

the text a comparison to hu.MAP 1.0. Specifically, we see hu.MAP 1.0 has function predictions for 197 uncharacterized proteins, compared to 274 in hu.MAP 2.0, an increase of ~40%. We also see hu.MAP 1.0 has 165 promiscuous proteins compared to 259 in hu.MAP 2.0, an increase of ~57%.

These comparisons are now included in our manuscript.

Major points

The differences in coverage as summarized above are not described in the manuscript (as it seems). Putting these or similar numbers in would help the reader judge the advances of hu.MAP 2.0.

See above.

The first sentence of the Results and Discussion section makes the following statement: "we can ask questions that were previously hindered by less accurate maps". To support this statement, the authors would need to show how the outcome of analyses improved using hu.MAP 2.0 compared to 1.0. For example, what is the increase in number of uncharacterized proteins for which hu.MAP 2.0 was able to make a function prediction compared to using hu.MAP 1.0? The same comparison could be done for the number of pleiotropic proteins identified.

We have softened our language in this section and have updated the text to compare our current map to our previous map as described above.

The validity of predicted pleiotropic proteins is evidenced by discussion of a single example, HSPA9. More systematic analyses seem more appropriate to validate the predictions as a whole. For example, the authors could try to show that the number of non-redundant GO terms for these proteins is higher than for the non-pleiotropic proteins that are also part of the identified protein complexes. The authors state the described example HSPA9 "demonstrates the ability of our complex map to identify multifunctional promiscuous proteins". This seems overstated without any additional systematic validations of the predictions.

We agree with the reviewer and have softened the language in the section to now read: "This example shows the ability of our complex map to identify multifunctional promiscuous proteins and place them into their respective non-overlapping functional complexes."

With respect to the systematic analysis recommended by the reviewer, we agree in principle this may show promiscuous proteins have more non-redundant functions than background but in practice GO is not sufficient to properly test this hypothesis. In particular, we used semantic similarity as a way to determine non-redundant functions where a low semantic similarity index represents two GO terms that are dissimilar and therefore likely nonredundant. We found that for the background set of non-promiscuous proteins which had > 2 GO Biological Process terms, 90% of the proteins had at least one pair of terms with a semantic similarity index < 0.1.

Taken at face value, this would mean that the background has an extremely high rate of nonredundant annotations counter to expectation. We believe the reason for this is that certain similar GO terms are on separate parts of the GO graph and therefore have low semantic similarity indices.

The authors identify that proteins that are part of multiple protein complexes in their dataset tend to be older/more conserved. I think in the HuRI paper, the authors showed that features like age and essentiality of genes correlate with their expression levels. Because MS has a well-known bias towards more highly expressed genes, the question arises whether the correlation between age and number of complex membership is confounded by an expression bias in the underlying MS-based datasets. Younger proteins might be less often detected in protein complexes because they are more lowly expressed and thus more often missed by MS. To support the observed trend between age and protein complex membership, it could be tested whether there is no significant trend between protein complex memberships and expression levels or at least, if such a trend was much weaker than the one observed between age and complex membership.

Please see response to Reviewer 1's point 7 and newly added Extended Figure 1 which shows negligible differences between expression levels of promiscuous and non-promiscuous proteins.

Does the performance assessment shown in Fig 2a assume that all interactions in a dataset that are not in CORUM are false positives? This is generally problematic but especially problematic for the HuRI dataset, which was generated using a very orthogonal method (Y2H) compared to MS-based techniques used in this manuscript. Y2H cannot find indirect associations, which dominate co-complex data (as evidenced in the HuRI paper), and more transient interactions that might be more easily detected by Y2H are usually lost in MS-based experiments due to washing procedures. While it is important to continuously remind readers about the technical and content-wise differences between Y2H-based and MS-based datasets, this analysis requires a more detailed interpretation compared to what is currently provided in the manuscript to avoid mis-interpretation by readers.

With respect to the test set of PPIs, pairs of proteins that are not in CORUM are unlabeled in our dataset. Pairs of proteins that are in CORUM but in separate complexes are labeled as "negative". We describe this in the methods section:

"A pair of proteins were labeled "positive" if both proteins were in the same complex. A pair was labeled "negative" if proteins were in separate complexes. All other pairs were left unlabeled."

With respect to Y2H, we have updated the text to avoid mis-interpretation by readers:

"Yeast 2-hybrid aims to capture only direct protein-protein interactions, and has previously shown good performance on benchmarks of binary interactors²³. Here we see HuRI underperforms all other networks when evaluated on co-complex interactions likely due to its inability to identify indirect physical interactions."

In the section entitled "Annotation enrichment" the authors describe the randomization of the protein complexes to calculate enrichments. From the brief description it remains unclear whether the number of complexes in which a protein participates was kept constant. This is key as proteins with a higher number of complex memberships are likely to be also better functionally annotated.

This is a very important point. To randomize protein complexes, we shuffled the links between protein ids and complex ids. In our original manuscript we mention that this has the effect of keeping both the number of complexes and size distribution of complexes the same as the original hu.MAP complexes. Our randomization method also keeps the distribution of complexes per protein the same and therefore we clarified the text to include this as well.

"In addition, this also has the effect of keeping the distribution of complexes per protein constant with the final hu.MAP 2.0 complexes as well."

The results page on the hu.MAP 2.0 web portal would benefit from providing information about the source datasets in which reported associations were identified. This would significantly help biologists to plan their experiments for the follow-up of identified interesting protein associations.

We wholeheartedly agree. The ability to trace each protein interaction back to its original experiment as well as the ability to determine if multiple orthogonal methods support an interaction is extremely helpful for users of the web resource to gain confidence in the identifications. Similar to our hu.MAP 1.0 web resource, we include in the "Edges" section of each complex a list of all interactions within the complex, each interaction's confidence score as reported by our machine learning pipeline, and the evidence for each interaction. The evidence column reports the datasets from which the interaction was derived and for datasets that have bait prey relationships, we list the bait gene name. Again, we feel this is an important aspect of all resources that integrate collections of datasets as it builds trust in the resource and as the reviewer mentioned allows users to better plan their next experiments.

Minor points

Individual figure panels should have individual letters and not 3 panels for example assigned to Figure 4b.

We updated the figures in Figure 4 to include individually labeled panels.

Figure 1b is referenced in the text to show that the WMM "provides evidence of interactions between many pairs of proteins not covered in the other datasets". How is this shown in this figure? This is an interesting point. It would be worth clarifying this simply by adding the count of unique interactions to the manuscript.

We thank the reviewer for this comment and agree it requires expanding. We now clarify in the figure legend to better describe the UpSet plot and state the total number of protein pairs which WMM provides additional information on (2.2×10^6).

The hu.MAP 2.0 web portal provides various datasets for download that are not provided as supplementary information in the manuscript. It would be helpful to the reader to point to the web portal in the manuscript where mentioned datasets are available for download, i.e. the 292 features.

Thank you for the suggestion. We now include links to additional downloads from our webresource.

CORUM asks users on their website to cite a more recent publication than the one cited in this manuscript, which seems appropriate given the more recent version of CORUM that has been used in this manuscript.

We now include the updated citation to CORUM.

The two if not the last 3 sections of the methods part might be better placed in the results part as they describe results of this study as also evidenced by the figures cited in these sections. The supplementary methods section contains a lot of information that is missing in the very general and sometimes uninformative description of the methods section. Moving some of this information from the supplement to the actual methods section would facilitate understanding of the methods by the reader.

We have now moved the supplementary methods into the main text as well as moved some of the previous methods sections into the results section.

To describe proteins with multiple different functions, the term pleiotropy is probably more commonly used than promiscuity that is used in this manuscript.

We considered the term pleiotropy as well as moonlighting but felt those terms had rigorous definitions in terms of gene and protein function. Here we show evidence of proteins interacting with multiple nonredundant complexes that is independent of function. We therefore felt the term promiscuous better represented the physical nature of the concept.

The authors hypothesize that "promiscuous proteins would be on average older due to younger proteins not having enough evolutionary time to make multiple connections". Is there any previous work that would support this idea and which could be cited?

We now include a citation to Saeed et al. 2006 which reports a strong correlation between protein age and protein interaction connectivity.

All examples picked to illustrate findings have evidence from Wan et al. It remains unclear whether the selected protein associations were also found in any of the other datasets? If yes, it might be worth adding this information to the manuscript text.

We now include references in the text to the evidence that supports the example complexes.

The description of the WMM in the methods section is somewhat confusing and actually much clearer in Hart et al. It might be helpful to explain what the authors mean by "experiment", i.e. one MS run of a pulldown or eluted fraction?

We have clarified the text describing the WMM in the methods section. In particular, we clarify an "experiment" is one pulldown experiment.

In equation 1, the index i is used. Where does i start? At 1?

Equation 1 defines $i=k\dots\min(n,m)$. We now explicitly state this in the text.

The authors describe that equation 1 has been calculated with different cutoffs for the different source datasets. Why?

Please see response to Reviewer 1 comment 14.

Reviewer #3:

The article "hu.MAP 2.0: Integration of over 15,000 proteomic experiments builds a global compendium of human multiprotein assemblies" by Drew et al presents an update on the hu.Map project, which the authors originally published in a 2017 MSB paper. The authors use their established computational pipeline from that paper to predict human protein complexes, by integrating various types of proteomics data from different sources. Features from the original articles are collected in a feature matrix together with a set of new features obtained by the authors' own WMM procedure, a SVM machine learning classifier is used to identify high-quality binary protein - protein interactions and a clustering algorithm identifies protein complexes. The resulting hu.Map 2.0 set of protein complexes appears to be the largest, and most accurate, set of computationally compiled protein complexes to date, and therefore presents an very valuable resource to the biomedical community.

To be clear, in contrast to the original hu.Map paper, this manuscript does not present anything new in terms of methods or approach, as far as I can tell. What is new here is that more datasets have been added and processed by the computational pipeline established by the authors. However, this significantly increased both accuracy and coverage of human protein complexes and therefore would be, in my opinion, a very useful publication from a resource perspective. For context, the authors' evaluation convincingly shows that huMap 2.0 clearly outperforms related resources, such as BioPlex (Huttlin et al, Cell 2015 and Nature 2017), which are in fact incorporated in huMap.

The pipeline itself is impressive and has been previously reviewed and validated. Therefore, I only have a few minor points that I think the authors should address:

- I find the title strongly misleading. The reference to 15,000 proteomics experiments sounds like the authors re-processed the MS raw files from scratch, as for example is done by the proteomicsDB project from the Kuester lab. This impression is reinforced by referring to input data as "raw features" and more or less explicitly stated in the sentence "To construct hu.MAP 2.0 we integrated over 15,000 previously published mass spectrometry experiments using our custom machine learning framework". However, here the authors download the processed tables and scored pairwise interactions from individual publications and use those as input for the machine learning, not the actual "raw" proteomics experiments. It is also not clear where the number 15,000 comes from (I assume this is the sum of MS raw files used by the combined set of input datasets).

In any case, this is not just an issue of semantics. While re-processing raw files from scratch may not be necessary, or even possible, for a dataset of this size, without doing so one cannot fully integrate the data either. This becomes an issue when analysing interactions between protein isoforms, for example when original articles mapped the same interaction to different isoforms because of the precise set of peptides observed in each experiment.

We thank the reviewer for this comment as it gives us an opportunity to clarify these concepts and we apologize for being unclear in our original manuscript. As we understand the reviewer brings up two points regarding first, the total number of experiments incorporated and second, the definition of "raw features". We originally listed the number of experiments from each technique in figure 1A but we now include in Table 1 the number of experiments from each individual dataset. With respect to the second point regarding "raw features", different fields consider "raw data" differently and we agree this is confusing language. We have therefore removed the term "raw" from our descriptions. We do not however consider there to be much of a difference between reprocessing the features starting from MS raw files and downloading the preprocessed features from the respective publications. Each dataset was produced by well respected and rigorous mass spectrometry labs and the processed features passed our internal quality control checks. We therefore did not expect to gain much from using a considerable amount of compute time to reprocess the data from scratch. The reviewer does mention isoforms which is something we do not deal with directly here. Most proteins identified in high throughput mass spec experiments are identified using just a handful of unique peptides and deciphering isoforms becomes a much more complicated task. It may be interesting for further work however. Finally, we feel it is important to portray to the reader the massive scale of mass spectrometry experiments that underlies this protein complex map. We therefore consider it appropriate to include in the title the total number of experiments integrated into the map as a way for the reader to judge the scale of evidence.

- For the PR curve, how does it look when you use just the WMM features?

While we did not explicitly create a classifier with just WMM features, we did create one lacking WMM features and evaluate its performance in Fig2A (labeled “No WMM”). From its performance, we see the WMM features are quite valuable when compared to the full hu.MAP 2.0 classifier. Specifically, we can look at the recall value at the 50% precision mark for both the full hu.MAP 2.0 classifier and the “No WMM” classifier and see a large drop in recall.

- Related to this, why is the performance of the HuRi data set so poor? Yeast 2 hybrid screens have been notorious for their high false positive rates in the past, but this striking apparent lack of performance could suggest that something else is going on. Could this reflect the fact that most of the interactions presented by huMap are in fact indirect interactions?

This is a point Reviewer 2 also noted and we addressed above but yes, the CORUM benchmark is made up of many co-complex (indirect) edges and the performance of the HuRi data is likely due to the inability of yeast 2-hybrid to detect indirect edges, as well as missing cases where other components of a multiprotein complex are required for proper assembly. An example of the latter is the COG complex, where HuRi records 79 PPIs from COG subunits to other proteins but none between COG subunits, presumably as the intact complex is required for their interactions to be properly captured. As we noted above in response to Reviewer 2, we have clarified the text to this point.

- On the subject of direct vs indirect interactions, can you estimate the relative fraction of each? I imagine that the huMap pipeline enriches for indirect interactions, due to the nature of the input data and using CORUM as a training set.

We note that the goal of our work is to define components of multiprotein complexes, not to distinguish contact surfaces within each complex. Thus, by design, our model classifies co-complex interactions, regardless of sharing surface interfaces, both due to the nature of the input data as well as the training data. In order to estimate the relative fraction of direct vs indirect interactions one would need to look to PDB structures of large complexes similar to our previous work (Drew et al. PLoSCompBio 2017), and this is the topic of future work.

- The biological follow-up and validation aspect is quite poor. Although the authors claim that the approach can be used to functionally characterise unknown proteins and identify the multiple functions of moonlighting proteins, there is no experimental demonstration of this claim, apart from highlighting a few previously known examples.

We understand the reviewer’s critique regarding limited additional experimental demonstration. We do however point out that the examples of functionally uncharacterized proteins are all new and point to new hypotheses to experimentally test. Additionally, our example of the uncharacterized protein CCDC9 as a member of the exon-exon junction complex is supported by independent experimental evidence in the form of differential fractionation experimentation (DIFFRAC, Mallam et al 2019). It is important to reiterate that the DIFFRAC data was not included to train the classifier and therefore is a valid independent experimental test of this observation.

- The statement "Unfortunately, we still lack a comprehensive set of protein complexes for the human cell" (page 2). Is there any evidence to support this statement, or are there any estimates about how many complexes actually exist? Or perhaps more relevantly, could one estimate when this approach will reach saturation? 7,000 "protein assemblies" seems to be a very large number already.

This is a very interesting point brought up by the reviewer. We have added a new section titled "Completeness of human protein complex map" to address this point.

"Our goal in this work is to build a complete and accurate set of protein complexes. We next asked how far have we come in achieving this goal? The size of the human interactome has previously been estimated to contain 154k-369k total interactions (Hart *et al*, 2006). Here we report 57k distinct interactions equating to roughly 15% - 37% complete. Consistent with this, if we consider the CORUM benchmark we use for evaluation in figure 2A as representative of all interactions, we see the majority of the hu.MAP 2.0 precision recall curve in figure 2A roughly falls between 15 and 37% recall. Also, making the same assumption for the CORUM benchmark, we can get rough estimates of hu.MAP 2.0's coverage of total protein complexes. The precision and recall metrics used in Figure 2D are robust to any redundancy and therefore give an accurate representation of total coverage. In Figure 2D, we see hu.MAP 2.0 covers >30% of complexes in the benchmark at a precision of 60%. As described in the section above, many cell-type and condition specific interactions likely make up a large portion of the remaining undiscovered interactions. We expect these interactions to be a focus of future experimentation in order to gain greater coverage of the complete human interactome."

- In the methods section, "Each individual experimental dataset identifies nonoverlapping sets of protein interactions" is misleading. They are hopefully partially overlapping.

We thank the reviewer for catching this unclear statement as did another reviewer. Please see Reviewer 1, comment 4 for our update.

- The "upset" plot is very useful and clearly better than a set of Venn Diagrams. However, it would be good to describe how it works in the legend, or provide a reference.

We now include a more detailed description of the UpSet plot in Figure 1B as well as a reference.

Thank you for sending us your revised manuscript. We have now heard back from the three reviewers who were asked to evaluate your study. The reviewers mention that the performed revisions have addressed most of their concerns and they are supportive of publication. Reviewer #1 would have appreciated a more detailed comparison of HuMAP2.0 with the dataset by Kustatscher et al. We agree that such a comparison would be informative. However, as reviewer #1 does not think that this is an essential point to address, we leave it up to you to decide whether to include it or not. Regarding the remaining issue raised by reviewer #3 (i.e. further clarifying that pre-processed and not raw data have been analysed), we would encourage you to further clarify this point as needed and mention the potential caveats.

On a more editorial level we would ask you to address the following remaining issues in a minor revision.

REFEREE REPORTS

Reviewer #1:

The authors have done an excellent job in revising their manuscript and have allayed most of my concerns regarding the first submitted version.
I would like to have seen a more detailed comparison of HuMAP2.0 with the Kustatcher et al data

set. They quote 'significant overlap' and an impressive p-value, but no detail - for example if there are any features between the two datasets that don't agree. I think that probably this is future work and could be omitted here, but I would be really keen to see where co-expression data and physical interaction data coincide and where they do not,

Reviewer #2:

I have no further comments or objections with respect to the publication of this manuscript.

Reviewer #3:

I find the revised version of the manuscript by Drew et al further improved and suitable for publication. I think it will be a very useful resource for the community.

Regarding my first revision point, the misleading title and overall impression that 15,000 MS raw files were directly mined for protein complexes: Removing the description as "raw" features is a step in the right direction, but the general problem persists. I agree it is important to convey to the reader that lots of data have been analyzed for this project, but there is a clear difference between looking at raw data and pre-processed data. In a way, it's the difference between reading the primary literature and a review article. The key information is most likely there, but you can't be absolutely certain about it and some information (like isoforms) is simply missing. It also means that expanding the approach (e.g. to include more data or new protein sequences) depends on other people pre-processing their data.

My second point about the "WMM only" PR curve was not addressed. This is not a major point.

The new section about the completeness of the human complex mapping is very interesting.

We again greatly appreciate the time and effort the reviewers put into reading our manuscript. We address each of the reviewers' specific points below.

Reviewer #1:

The authors have done an excellent job in revising their manuscript and have allayed most of my concerns regarding the first submitted version.

I would like to have seen a more detailed comparison of HuMAP2.0 with the Kustatcher et al data set. They quote 'significant overlap' and an impressive p-value, but no detail - for example if there are any features between the two datasets that don't agree. I think that probably this is future work and could be omitted here, but I would be really keen to see where co-expression data and physical interaction data coincide and where they do not,

We thank the reviewer for their insightful comments. We share the reviewer's interest in comparing the co-expression data and our complex map and believe there is likely more to be discovered in regard to this analysis. Unfortunately, in order to draw any conclusions stronger than what we have already done, we believe would require a more sophisticated statistical approach that is outside of the scope of the current work. For example, due to the high false negative rates of the datasets, it would be difficult to determine whether the co-expression data and physical interaction data do not coincide because of a principle of biological significance or due to biases in the collection of the data. We therefore leave this analysis for future work.

Reviewer #2:

I have no further comments or objections with respect to the publication of this manuscript.

We thank the reviewer for their original comments.

Reviewer #3:

I find the revised version of the manuscript by Drew et al further improved and suitable for publication. I think it will be a very useful resource for the community.

We thank the reviewer for their comments as we feel they improved our manuscript.

Regarding my first revision point, the misleading title and overall impression that 15,000 MS raw files were directly mined for protein complexes: Removing the description as "raw" features is a step in the right direction, but the general problem persists. I agree it is important to convey to the reader that lots of data have been analyzed for this project, but there is a clear difference between looking at raw data and pre-processed data. In a way, it's the difference between reading the primary literature and a review article. The key information is most likely there, but you can't be absolutely certain about it and some information (like isoforms) is simply missing. It also means that expanding the approach (e.g. to include more data or new protein sequences) depends on other people pre-processing their data.

We respectfully disagree with the reviewer on this point. The reviewer suggests there would be a difference between starting our analysis with the raw mass spectra versus the processed data. If we were to start with raw spectra, we would have applied similar or even identical software pipelines to obtain very similar processed results. In addition, as we pointed out in our

initial response, the data we used in our analysis were published by well-respected and rigorous mass spectrometry labs. We felt it would be highly unlikely that we would substantially improve on their processing pipelines and therefore results. We agree with the reviewer that some information like isoforms is missing but although interesting, this was not the focus of our work and we would have ignored isoforms if we reprocessed the data ourselves. We also disagree with the reviewer's last point that our approach depends on other people pre-processing their data. The framework is agnostic about who processes the raw mass spec data. As an example, we reprocessed the Treiber et al. dataset (originally for Mallam et al. 2019) and included it in our analysis. It should be noted our reanalysis was because the published processed data did not include the required information rather than any error in the processing of the published data.

In short, we built on the work of others. We feel this is an important aspect of the mass spectrometry community which makes publicly available both raw and processed data in order for other groups to build upon and ultimately further the field.

My second point about the "WMM only" PR curve was not addressed. This is not a major point.

The reviewer is correct that we did not explicitly create a WMM only model and evaluate its PR curve but as we explained in our original response we did create a model that lacked WMM features providing the ability to evaluate the substantial gain in performance when WMM features were included. We felt this analysis addressed the reviewer's question.

The new section about the completeness of the human complex mapping is very interesting.

We share the reviewer's interest in this analysis and appreciate their original comment posing the question about completeness of our human complex map.

Thank you again for sending us your revised manuscript. We are now satisfied with the modifications made and I am pleased to inform you that your paper has been accepted for publication.

Corresponding Author Name: Edward Marcotte

Manuscript Number: MSB-20-10016